# HIERARCHICAL MULTIMODAL VARIATIONAL AUTOENCODERS

## ABSTRACT

Humans find structure in natural phenomena by absorbing stimuli from multiple input sources such as vision, text, and speech. We study the use of deep generative models that generate multimodal data from latent representations. Existing approaches generate samples using a single shared latent variable, sometimes with marginally independent latent variables, to capture modality-specific variations. However, there are cases where modality-specific variations depend on the kind of structure shared across modalities. To capture such heterogeneity, we propose a hierarchical multimodal VAE (HMVAE) that represents modality-specific variations using latent variables *dependent* on a shared top-level variable. Our experiments on the CUB and the Oxford Flower datasets show that the HMVAE can outperform existing methods in terms of generative heterogeneity and coherence across several quantitative and qualitative measures. We provide the code to reproduce the results in the supplementary material.

## 1 INTRODUCTION

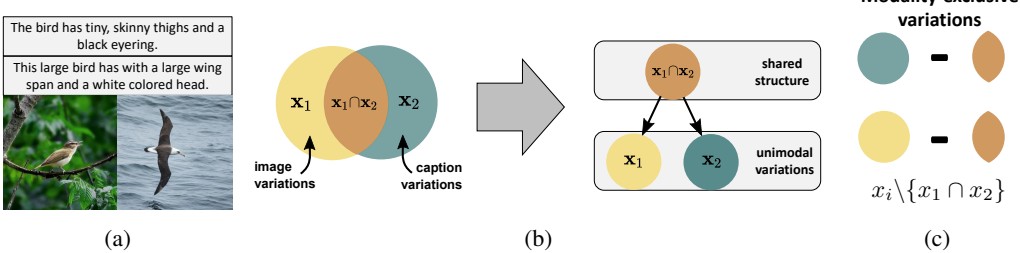

Figure 1: **Hierarchical decomposition.** (a) The image background in the two pictures depends on the bird species. (b) We propose to capture such dependencies via hierarchies. (c) A closer look at the image background reveals that it is modality-exclusive, i.e., not described by the caption.

Data modalities represent different perspectives of the same concept. Generative models can learn from such data by reproducing it, which can be useful for tasks such as image caption generation (Vinyals et al., 2015). This model family can also reproduce feature vector representations of the data, which can be helpful for tasks such as zero-shot classification (Xian et al., 2018b) or reinforcement learning (Bruce et al., 2017). One difficulty in multimodal learning lies in the differing probabilistic structures across modalities (Fig. 1). Our goal is to develop generative models that capture unimodal variations in both modalities.

One line of previous works (Zhu et al., 2017; Zhang et al., 2017; Reed et al., 2016) tackled this challenge with conditional generative adversarial networks (GANs) (Mirza and Osindero, 2014). These models generate samples for one modality conditioned on another modality and are optimized via competition between a generator and a discriminator. In contrast, Suzuki et al. (2016) used variational autoencoders (VAEs) to jointly generate $M$ modalities from a learned latent representation. We focus on VAEs, which are explicit density estimators and can maximize the likelihood of all data variations. GANs are implicit estimators, which can cause the generator to disproportionally favor specific variations (Razavi et al., 2019).

Some multimodal VAEs (Wu and Goodman, 2018; Shi et al., 2019) incorporate a single latent variable $g$ that captures all relevant information (Fig. 2a). This formulation may disregard modality-exclusive variations. Other works (Huang et al., 2018; Hsu and Glass, 2018; Mahajan et al., 2020; Sutter et al., 2020; Daunhawer et al., 2021b; Lee and Pavlovic, 2021) have introduced disentangled latent variables $z_{1:M}$, which are marginally independent of a shared latent variable $g$ and represent structure specific to modality $i$ (Fig. 2b). We argue that a disentangled latent representation may neglect the dependencies between shared and modality-exclusive variations.

As a specific example, consider captioning (modality 2) pictures of birds (modality 1) as in Fig. 1a. The images of seabirds can have sky or water in the background, while those of songbirds often display forest backgrounds. The captions focus on the bird and thereby easily ignore such variations. Therefore, the shared pattern across modalities (bird species) dictates these modality-exclusive variations. Consider a generative model where $g$ represents shared structure and $z_1$ image variations. When the two latent variables $g$ and $z_1$ are independent (Fig. 2b), the decoder that maps from latent variables onto images has to learn two distinct functions - one for seabirds (where the independent variations determine the sky or water background) and another for songbirds (where the independent variations determine the forest background). We argue that this aspiration is theoretically learnable but practically challenging because it may require a large model with disproportional capacity (that could generalize poorly, is challenging to train, or requires abundant data). In contrast, we suggest that a hierarchical latent representation is an inductive bias that captures realistic data variations and guides learning. The edges between $g$ and $z_{1:M}$ give the model the flexibility to decide which unimodal variations to capture given a shared latent concept. For example, a hierarchical model could *adaptively* learn that $z_1$ for seabirds solely captures sky or water background variations while $z_1$ for songbirds captures forest background variations. This can help the decoder to share features between seabirds and songbirds.

**Contributions** **(i)** We propose a hierarchical multimodal VAE (HMVAE) that incorporates a latent hierarchy, where the shared variable $g$ resides at the top and lower variables $z$ complement unimodal variations. **(ii)** We compare the HMVAE to several state-of-the-art baselines on the CUB and the Oxford Flower datasets. We report improved quantitative and qualitative measures in terms of semantic coherence and heterogeneity.

## 2 BACKGROUND AND RELATED WORK

**Variational Autoencoders (VAEs)** (Kingma and Welling, 2013; Rezende et al., 2014) are deep generative models that represent the joint distribution $p_\theta(x, z)$ using neural networks with parameters $\theta$, where $x \in \mathbb{R}^D$ is the observed vector and $z \in \mathbb{R}^{D'}$ is the latent vector. As the true posterior $p_\theta(z|x)$ is intractable, an approximate posterior $q_\phi(z|x)$ with parameters $\phi$ is used for inference. The parameters $\theta, \phi$ are usually trained by maximizing the evidence lower bound (ELBO) for the marginal likelihood:

$$\mathbb{E}_{q_\phi(z|x)} \left[ \log \frac{p_\theta(x,z)}{q_\phi(z|x)} \right] \leq \log p_\theta(x). \tag{1}$$

Many efforts have been made to increase the expressivity of VAEs, e.g., by improving the prior of $z$ (Chen et al., 2016; Tomczak and Welling, 2018), and by introducing auxiliary latent variables (Maaløe et al., 2016). Our approach lies in the latter paradigm.

**Hierarchical VAEs (HVAEs)** (Rezende et al., 2014) have a hierarchical latent structure, where the topmost latent variable, drawn from an unconditional prior $p_\theta(z_L)$, represents global features. The lower variables, drawn from conditional priors $p_\theta(z_i|z_{i+1})$, complement local characteristics in order to reconstruct the observed data via $p_\theta(x|z_1)$.

Sønderby et al. (2016) found the tendency for HVAEs to not effectively use higher-level latent variables when they are trained using inference networks of the form $q_\phi(z_{i+1}|z_i)$. They proposed to first infer the top-level variable $z_L$ directly with $q_\phi(z_L|x)$ and then infer the intermediate variables $\{z_i\}$ with both bottom-up and top-down information through $q_{\phi,\theta}(z_i|z_{i+1}, x)$ for hierarchical level $i \in \{0, ..., L-1\}$. The bottom-up and top-down information for hierarchical level $i$ are encoded as hidden variables through neural networks: $b_i = f_{\phi,i}(x)$ and $t_i = f_{\phi,\theta,i}(z_{i+1})$. We make use of this idea in our work, concatenate both hidden states and pass the result to an MLP that parameterizes the respective posterior (see App. C.1 for further details). This inference procedure can result in im-

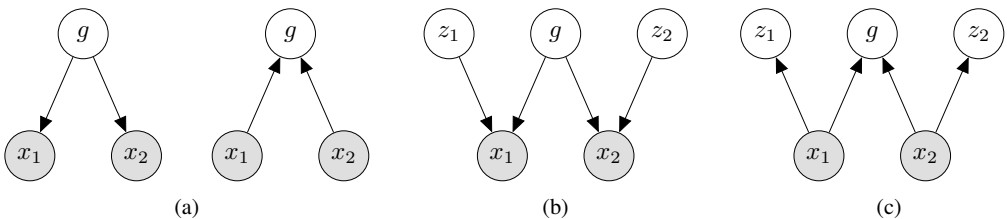

Figure 2: **Related work.** (a) MVAE, MMVAE, (b) MDVAE, (c) MDVAE, MHVAE. Note that we discuss an alternative inference network for the MDVAE in App. D.

proved density estimation and sample generation performance (Sønderby et al., 2016; Maaløe et al., 2019; Vahdat and Kautz, 2020; Child, 2021).

**Multimodal VAEs** represent $M$ modalities $\boldsymbol{x}_{1:M} = \{\boldsymbol{x}_1, ..., \boldsymbol{x}_M\}$ that are assumed to be conditionally independent given a shared representation $\boldsymbol{g}$: $p_\theta(\boldsymbol{x}_{1:M}|\boldsymbol{g}) = \prod_{m=1}^{M} p_\theta(\boldsymbol{x}_i|\boldsymbol{g})$. Motivated by the factorization of the true posterior, Wu and Goodman (2018) used a product of experts (PoE) formulation to parameterize the approximate posterior distribution over the shared latent variable, where each expert characterizes information within a modality:

$$q_\phi(\boldsymbol{g}|\boldsymbol{x}_{1:M}) \propto p_\theta(\boldsymbol{g}) \textstyle\prod_{m=1}^{M} q_\phi(\boldsymbol{g}|\boldsymbol{x}_m). \tag{2}$$

When all experts are Gaussian, the product posterior of any combination of experts is easily computed.

Shi et al. (2019) showed that an inference model with a Gaussian PoE formulation can be miscalibrated, i.e., overconfident experts $q_\phi(\boldsymbol{g}|\boldsymbol{x}_i)$ with low densities dominate the product $q_\phi(\boldsymbol{g}|\boldsymbol{x}_{1:M})$. They instead used a mixture of experts (MoE) formulation:

$$q_\phi(\boldsymbol{g}|\boldsymbol{x}_{1:M}) = \tfrac{1}{M} \textstyle\sum_{m=1}^{M} q_\phi(\boldsymbol{g}|\boldsymbol{x}_m). \tag{3}$$

With the MoE formulation, optimization is analogous to a vote, which can avoid unreasonable dominance by a single modality.

Mahajan et al. (2020) applied normalizing flows (Rezende and Mohamed, 2015) to map between the latent spaces of $\boldsymbol{g}_1$ and $\boldsymbol{g}_2$ in a VAE, where $q_\phi(\boldsymbol{g}_i|\boldsymbol{x}_i)$ constitutes the posterior over $\boldsymbol{g}_i$. Normalizing flows can be more expressive than other distribution choices, such as Gaussians, which can help to represent complex multimodal relationships. However, optimizing the ELBO requires tractable sampling and density estimation in *either* direction, for example, by using coupling layers (Dinh et al., 2014; 2017). This can be a challenging constraint in practice.

Vasco et al. (2020) proposed the MHVAE, which also incorporates a hierarchical generative model. Note that both works were developed independently and offer complementary perspectives. We summarize the differences in the following: first, the MHVAE's generative model is limited to two hierarchical levels (like Fig. 3a, but for two levels). Our proposed generative model supports arbitrary hierarchical depth. Second, the MHVAE's inference model is non-hierarchical (Fig. 2c, Eq. 9 from the original paper). Our proposed inference model is hierarchical (Fig. 3b) Third, the MHVAE's posteriors over $\boldsymbol{z}$ are unimodal. Our proposed *"top-down"* inference model computes the latent variables in the same order as the generative model. Therefore, the respective posteriors depend on all modalities. Fourth, the MHVAE's inference network drops modality-specific hidden states at random during training. We use a mixture of experts posterior. Fifth, Vasco et al. (2020) evaluate their model on surjective data where single labels or attributes describe classes of images, i.e., there is not much variation in a single modality but lots in the other. We focus on a different data scenario where each modality has a large degree of variation.

## 3 MULTIMODAL LATENT HIERARCHIES

We propose a hierarchical multimodal VAE (HMVAE) that captures the generative process of multiple modalities. The hyperparameter $L(m) \geq 2$ defines the number of hierarchical levels for modality $m$. If $L(m) = 2$, the hierarchy solely expands between the shared and unimodal variables. If $L(m) > 2$, the model incorporates additional unimodal hierarchies.

**Generative model** We assume conditional independence of modalities given a shared variable $\boldsymbol{g}$ (see Fig. 3a portraying the case of two modalities and three hierarchical levels):

$$p_{\phi,\theta}(\boldsymbol{x},\boldsymbol{g},\boldsymbol{z}) = \prod_{m=1}^{M} p_{\theta}(\boldsymbol{x}_m|\boldsymbol{z}_{m,1})$$
$$\cdot \left( \prod_{i=1}^{L(m)-2} p_{\phi,\theta}(\boldsymbol{z}_{m,i}|\boldsymbol{z}_{m,i+1}) \right) \quad (4)$$
$$p_{\phi,\theta}(\boldsymbol{z}_{m,L(m)-1}|\boldsymbol{g}) \; p_{\theta}(\boldsymbol{g}).$$

The prior for the shared variable $\boldsymbol{g}$ is isotropic Gaussian. All conditional distributions for the intermediate variables $\{\boldsymbol{z}_{m,1:L(m)-1}\}$ for $m \in \{1,\dots M\}$ are also isotropic Gaussian, where mean and variance are parameterized using neural networks. We use the hierarchical formulation introduced in § 2 where some parameters are shared across the generative and inference networks. The conditional distributions for the observed variables $\{\boldsymbol{x}_m\}$ can, for example, be parameterized using isotropic Gaussian distributions (for continuous-valued data) or Bernoulli distributions (for binary data).

**Inference model** We extend Sønderby et al. (2016)'s hierarchical approach described in § 2 to the multimodal case (see Fig. 3a depicting the case of two modalities and three hierarchical levels):

Figure 3: **Proposed generation and inference.** The HMVAE represents unimodal variations in conditional variables $\boldsymbol{z}$. We generalize the inference network from Sønderby et al. (2016) to the multimodal case, where all posteriors are multimodal. For example, the red edges visualize how an observed modality $\boldsymbol{x}_2$ can affect $\boldsymbol{z}_{1,1}$ in the other modality. The figure visualizes the special case of three hierarchical levels.

$$q_{\phi,\theta}(\boldsymbol{g},\boldsymbol{z}|\boldsymbol{x}_{1:M}) = q_{\phi}(\boldsymbol{g}|\boldsymbol{x}_{1:M}) \cdot \prod_{m=1}^{M} q_{\phi,\theta}(\boldsymbol{z}_{m,L(m)-1}|\boldsymbol{g},\boldsymbol{x}_m) \prod_{i=1}^{L(m)-2} q_{\phi,\theta}(\boldsymbol{z}_{m,i}|\boldsymbol{z}_{m,i+1},\boldsymbol{x}_m). \quad (5)$$

All distributions except the first factor $q_{\phi}(\boldsymbol{g}|\boldsymbol{x}_{1:M})$ are isotropic Gaussian with mean and variance inferred through neural networks from the conditional variables. The network employs skip connections from $\boldsymbol{x}_{1:M}$ to $\boldsymbol{g}$. The inference and generative networks share some parameters in the top-down networks that point from $\boldsymbol{g}$ to the lowest unimodal variable $\boldsymbol{z}_{m,i=0}$. This architecture can reinforce hierarchical decomposition (Sønderby et al., 2016). Furthermore, it ensures that the unimodal latent variables are conditioned on *all* modalities (as indicated by the red edges in Fig. 3b) which helps learn crossmodal relationships. Section 2 and App. C.1 provide further details on the hierarchical architecture.

We parameterize the posterior distribution over the shared latent variable $q_{\phi}(\boldsymbol{g}|\boldsymbol{x}_{1:M})$ using a mixture of experts formulation (Eq. 3). We follow Shi et al. (2019) and assume that several modalities can entail modality-exclusive variations, i.e., some variations for modality $i$ do not correlate with variations in modality $j \neq i$.

**Optimization** We train the HMVAE by maximizing the ELBO with stochastic backpropagation:

$$ELBO := \mathbb{E}_{q_{\phi,\theta}(\boldsymbol{g},\boldsymbol{z}|\boldsymbol{x}_{1:M})} \left[ \log \frac{p_{\phi,\theta}(\boldsymbol{x}_{1:M},\boldsymbol{g},\boldsymbol{z})}{q_{\phi,\theta}(\boldsymbol{g},\boldsymbol{z}|\boldsymbol{x}_{1:M})} \right] \leq \log p_{\theta}(\boldsymbol{x}_{1:M}). \quad (6)$$

**Further motivation** Figure 4 visualizes how multimodal VAEs could capture the data from the example in the Introduction (Fig. 1). The challenge lies in representing crossmodal dependencies – not the joint distribution. That is because one modality can never inform about the modality-exclusive variations of another modality. Daunhawer et al. (2021a) demonstrated that this problem constitutes a core limitation across the multimodal VAE literature. In a non-hierarchical VAE, crossmodal generation is indeed challenging because $\boldsymbol{g}$ must capture all variations. This becomes problematic when generating $p_{\theta}(\boldsymbol{x}_i|\boldsymbol{g})$ from $q_{\phi}(\boldsymbol{g}|\boldsymbol{x}_{j \neq i})$ because the latter misses information about $\boldsymbol{x}_i$. In contrast, in a two-level hierarchy, $\boldsymbol{z}$ could theoretically capture the entirety of modality-exclusive variations. However, the model may choose to represent some of these variations in $\boldsymbol{g}$ to capture the hierarchical dependencies *within* the modality-exclusive variations. A deep hierarchy could circumvent this

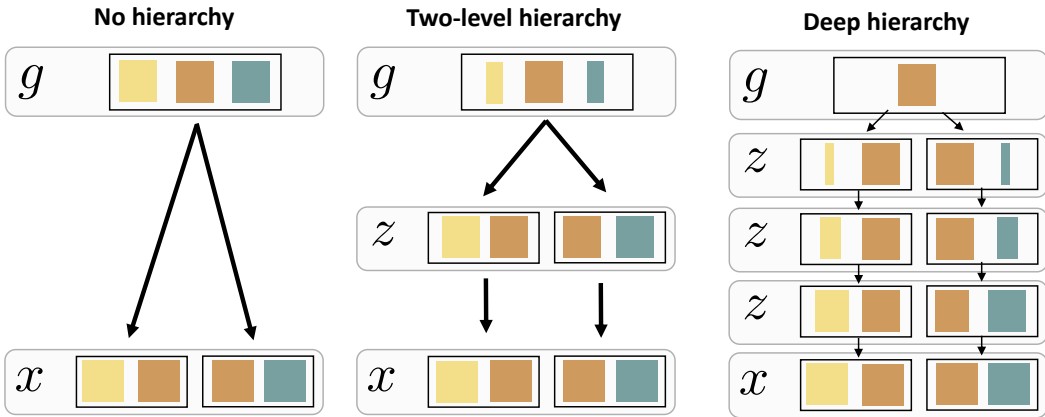

Figure 4: **Hierarchical models can factorize unimodal variations along the hierarchy** This figure visualizes possible latent representations for the multimodal data from Fig. 1: two modalities $x_1$ and $x_2$, where the brown color indicates shared structure and the yellow/green colors modality-exclusive variations.

problem by adding infinitesimal small modality-exclusive variations per hierarchical level (in the limit). Note that Fig. 4 demonstrates one set of possible hierarchical representations. A model could also utilize its degrees of freedom differently. For example, the deep model may incorporate shared structure *and* modality-exclusive variations for the more complex modality in $g$. In general, we expect a (deep) hierarchy to be beneficial for representing complex data such as image or text modalities.

## 4 EXPERIMENTS

We evaluate on the *CUB* (Wah et al., 2011) and *Oxford Flower* (Nilsback and Zisserman, 2008) datasets. The modalities in both datasets are images and captions (text). We will focus on cross-modal generation, i.e., caption-to-image generation and the reverse. We also consider unconditional generation. We evaluate generated samples mainly in terms of heterogeneity and coherence. Heterogeneity describes diversity across samples for a single setting of the shared latent variable g and is essential to ensure that the generated samples capture the diversity of the data. Coherence indicates whether a crossmodal sample lies on the manifold of the true data. Crucially, heterogeneity and coherence are often both necessary. Imagine a model that produces class-agnostic generations (high heterogeneity, low coherence for classification tasks). As for the HMVAE, we will validate that the model hierarchically decomposes the data and outperforms state-of-the-art competitors.

### 4.1 EXPERIMENTAL DESIGN

**Models** We are not aware of any standard benchmark dataset in the literature with reported performance for all considered baselines and sufficient hierarchical structure in the data. Therefore, we reimplement all considered baselines: the MVAE (Wu and Goodman, 2018), the MMVAE (Shi et al., 2019), the MDVAE (Huang et al., 2018; Hsu and Glass, 2018; Mahajan et al., 2020; Sutter et al., 2020; Daunhawer et al., 2021b; Lee and Pavlovic, 2021), and the HMVAE (Vasco et al., 2020). Related works suggested many different MDVAEs variants. Our reimplementation uses a posterior choice suitable to the given data regime and stays close to the HMVAE– except that $z$ are marginally independent of $g$. In general, we keep the architecture and optimization hyperparameters similar across all models. App. C describes our implementation in detail. The use of larger models may produce photorealistic images with a high computational cost. For example, Child (2021) train a unimodal hierarchical VAE for around 2.5 weeks using 32 *V100* GPUs on datasets such as *FFHQ*. However, our goal is not photorealism but to validate that the proposed HMVAE represents multimodal data differently than the baselines.

**Data** In some of our experiments, we follow the common practice in the multimodal learning literature and preprocess the data by extracting feature vector representations (Xian et al., 2018b; Sariyildiz and Cinbis, 2019; Schonfeld et al., 2019; Shi et al., 2019; 2020). Across all experiments, we use the caption feature vectors provided by Zhang et al. (2017). These feature vectors $x_2 \in \mathbb{R}^{1024}$ were extracted using a CNN-RNN (Reed et al., 2016). Each image is paired with ten captions. We average the caption features for each image (except for the qualitative experiments which visualize captions explicitly). We find that this procedure can improve training and reduce computational requirements. In § 4.2, we extract features $x_1 \in \mathbb{R}^{2048}$ from a ResNet-101 trained on ImageNet. In § 4.3, we use the images $x_1 \in \mathbb{R}^{3 \times 64 \times 64}$ directly. We refer to App. C for details. To visualize image features and caption features, we follow Shi et al. (2019; 2020), search for the nearest neighbor feature vector in the test set using the mean from $p(x_i|\cdot)$, and visualize the respective image or caption. Using this practice, we can evaluate coherence (relationship between generation and condition) and heterogeneity (sample diversity). To improve readability, we trim exceptionally long captions in the qualitative figures.

## 4.2 FEATURE GENERATION

In this section, we maximize the likelihood of image features $x_1 \in \mathbb{R}^{2048}$ and caption features $x_2 \in \mathbb{R}^{1024}$ on the CUB and the Oxford Flower datasets. We will validate the general utility of latent hierarchies relative to single-variable models. For the HMVAE, we choose $L(1) = L(2) = 2$.

Figures 5 and 6 display $p(x_1|x_2)$ and $p(x_2|x_1)$, respectively. For the baselines, we vary $g$ to generate $x_i$, where $g^{1:K} \sim q(g|x_{j\neq i})$ and $p(x_i|g^k)$. For the proposed HMVAE, we utilize the hierarchy by varying $z_i$ to generate $x_i$, where $g \sim q(g|x_{j\neq i})$, $z_i^{1:K} \sim p(z_i|g)$, and $p(x_i|z_i^k)$. All models generate features that are mainly semantically coherent with the condition, indicating high coherence. However, the proposed HMVAE generates samples of higher diversity than the baseline, indicating high heterogeneity. This diversity must be represented in $z_1$, because we fix $g$ and vary $z_1$. Table 1 demonstrates that the HMVAE achieves state-of-the-art performance across most evaluated likelihoods. Likelihood estimates quantify both coherence and heterogeneity because they measure the divergence between the true and approximated distribution.

Table 1: **Likelihood estimates.** We approximate the joint, marginal, and crossmodal likelihoods (defined in App. B) of test samples using 500 importance weighted samples. All models maximize the likelihood of image feature vectors and caption feature vectors. We report average and standard deviation over five runs per model. (higher is better)

| Dataset and Model | $\log p(x_1, x_2)$ | $\log p(x_1)$ | $\log p(x_2)$ | $\log p(x_1|x_2)$ | $\log p(x_2|x_1)$ |
|---|---|---|---|---|---|
| **CUB** | | | | | |
| MVAE (Wu and Goodman, 2018) | -2046.0 $\pm$ 6.6 | -1902.3 $\pm$ 8.8 | 142.8 $\pm$ 16.1 | -3594.0 $\pm$ 337.0 | -3024.9 $\pm$ 297.4 |
| MMVAE (Shi et al., 2019) | -2633.5 $\pm$ 47.7 | -2075.5 $\pm$ 8.2 | 180.9 $\pm$ 10.5 | -3227.2 $\pm$ 145.1 | -2022.2 $\pm$ 77.5 |
| HMVAE (**this work**) | **-1424.8 $\pm$ 14.8** | **-1854.8 $\pm$ 14.2** | **439.6 $\pm$ 2.0** | **-2382.9 $\pm$ 19.9** | **-879.7 $\pm$ 31.8** |
| | | | | | |
| **Oxford Flower** | | | | | |
| MVAE (Wu and Goodman, 2018) | -2471.3 $\pm$ 59.4 | **-2433.4 $\pm$ 22.5** | 215.4 $\pm$ 21.9 | -3067.7 $\pm$ 67.5 | -5282.8 $\pm$ 835.1 |
| MMVAE (Shi et al., 2019) | -3029.9 $\pm$ 50.5 | -2575.5 $\pm$ 8.6 | 141.2 $\pm$ 26.0 | -3179.3 $\pm$ 44.2 | -3454.0 $\pm$ 218.4 |
| HMVAE (**this work**) | **-2069.1 $\pm$ 40.0** | -2497.6 $\pm$ 21.7 | **442.2 $\pm$ 27.7** | **-3009.3 $\pm$ 38.0** | **-1400.0 $\pm$ 77.1** |

## 4.3 IMAGE GENERATION

In this section, we maximize the likelihood of images $x_1 \in \mathbb{R}^{3 \times 64 \times 64}$ and caption feature vectors $x_2 \in \mathbb{R}^{1024}$. Because the data is more complex than in the previous section, we now investigate deeper hierarchies for the HMVAE[1] and incorporate additional baselines with multiple latent variables (MDVAE and MHVAE). Note that the deep HMVAE employs the fewest parameters (see Table 4 for an overview) and is the only model that does not use a warmup scheme for the KL-divergence loss (Sønderby et al., 2016) .

**Quantitative results** The *Fréchet Inception Distance (FID)* compares the means and covariances between the true and approximate distributions in the feature space of an Inception network (Heusel

---

[1]The shallow variant incorporates $L(1) = L(2) = 2$, the deeper variant incorporates $L(1) = 5$ and $L(2) = 2$.

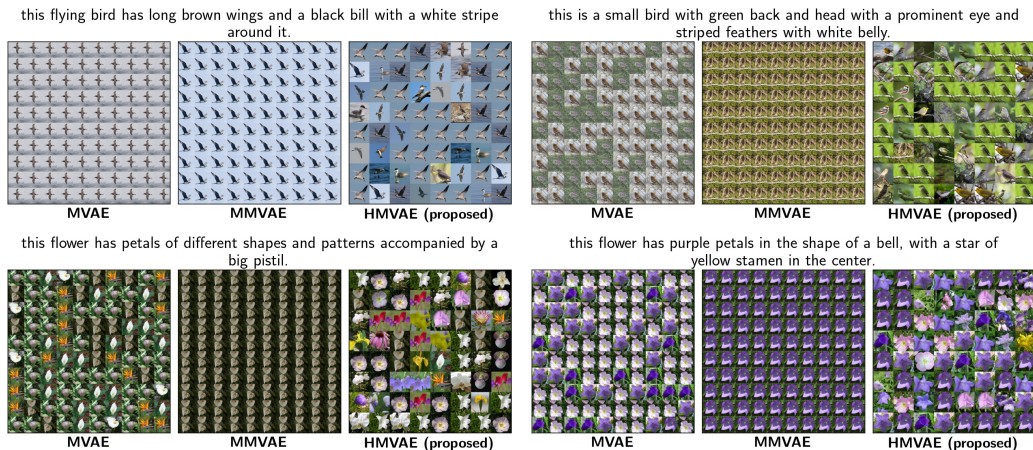

Figure 5: **CUB/Oxford Flower: Generating image feature vectors from caption feature vectors.**
We generate image features, look up the nearest-neighbor feature in the test set, and visualize the associated image.

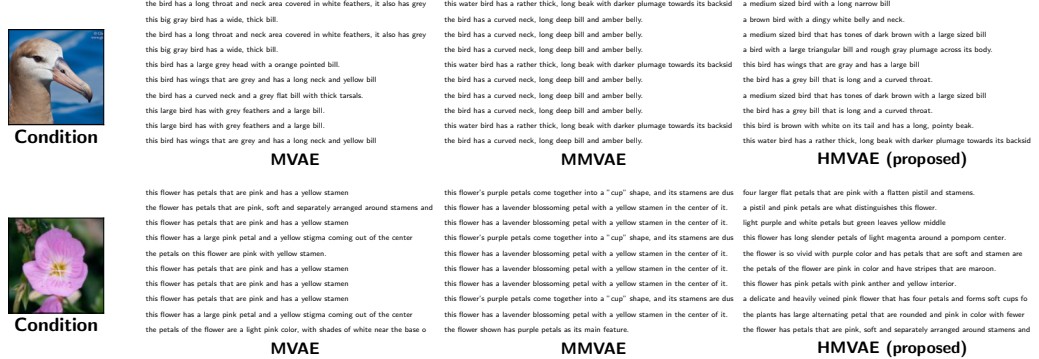

Figure 6: **CUB/Oxford Flower: Generating caption feature vectors from image feature vectors.**
We generate caption features, look up the nearest-neighbor feature from the test set, and visualize the respective caption.

et al., 2017; Seitzer, 2020). For the remaining measures, we preprocess images by extracting feature representations from a ResNet-101 pre-trained on ImageNet. For example, we generate images $\boldsymbol{x}_1 \in \mathbb{R}^{3 \times 64 \times 64}$ and then project these samples into the mentioned feature space. We do not use any preprocessing for the caption features because of their lower dimensionality (1024). Precision measures the relative amount of generated samples that lie on the true manifold. Recall measures the opposite. We follow (Kynkäänniemi et al., 2019) and approximate the target manifold using hyperspheres with radii equivalent to the 3-nearest-neighbors of the respective samples. We also compute the harmonic mean across precision and recall (F1). Finally, we compute the average variance of image representations: for each conditioning caption, we sample $\boldsymbol{g}^{1:100} \sim q(\boldsymbol{g}|\boldsymbol{x}_2)$ to generate $p(\boldsymbol{x}_1|\boldsymbol{g})$ and project these samples into the ResNet feature space $f(\boldsymbol{x}_1|\boldsymbol{g})$. We then compute the variance of these samples to a prototype representation $f(\bar{\boldsymbol{x}_2}|\boldsymbol{g}) = \frac{1}{100}\sum_{i=1}^{100} f(\boldsymbol{x}_2^i|\boldsymbol{g})$.

Table 2a presents quantitative results.[2] The MHVAE and the HMVAE are the only models that perform reasonably well across all evaluated measures. For example, even though the MVAE achieves the best FID, the model scores poorly in the precision measures. Furthermore, the model's qualitative samples display a lack of fidelity (Fig. 7). In contrast, the MMVAE generates coherent samples (high precision) but lacks heterogeneity (low variance and recall). The HMVAE also uses a mixture

---

[2]Note that we report the image recall and F1 values for the sake of completeness. These values are low across all models due to the complexity of the image space given a relatively small dataset.

Table 2: **Oxford Flower** All models maximize the likelihood of images and caption features vectors. We report average and standard deviation over three runs per model. (a) The second column highlights the number of latent variables in the model. (b) Hierarchical decomposition in the HMVAE.

(a)

| Model | # LV | $p(\boldsymbol{x}_1)$ vs. $p(\boldsymbol{x}_1\|\boldsymbol{x}_2)$ | | | | | $p(\boldsymbol{x}_2)$ vs. $p(\boldsymbol{x}_2\|\boldsymbol{x}_1)$ | | |
|---|---|---|---|---|---|---|---|---|---|
| | | FID | Precision | Recall | F1 | Variance | Precision | Recall | F1 |
| MVAE (Wu and Goodman, 2018) | 1 | **142.1 ± 6.2** | 75.8 ± 1.3 | 2.9 ± 0.5 | **5.5 ± 0.9** | 0.51 ± 0.02 | 7.1 ± 4.6 | 92.8 ± 3.0 | 12.7 ± 7.9 |
| MMVAE (Shi et al., 2019) | 1 | 161.1 ± 4.5 | 91.5 ± 2.3 | 1.3 ± 0.1 | 2.6 ± 0.3 | 0.04 ± 0.0 | 74.3 ± 1.0 | 26.1 ± 1.8 | 38.6 ± 1.8 |
| MDVAE (Mahajan et al., 2020), i.a. | 3 | 167.8 ± 3.1 | 91.4 ± 1.9 | 0.9 ± 0.4 | 1.7 ± 0.8 | 0.22 ± 0.0 | 6.1 ± 3.2 | 53.7 ± 1.0 | 10.7 ± 5.0 |
| HMVAE (ablation) | 3 | 162.6 ± 2.2 | 95.2 ± 1.0 | 1.3 ± 0.2 | 2.5 ± 0.3 | 0.12 ± 0.01 | 81.8 ± 0.9 | 20.4 ± 2.0 | 32.6 ± 2.5 |
| MHVAE (Vasco et al., 2020) | 3 | 159.9 ± 3.2 | 86.8 ± 1.2 | 1.6 ± 0.2 | 3.2 ± 0.4 | 0.29 ± 0.01 | 53.4 ± 3.0 | 63.0 ± 4.2 | 57.7 ± 2.6 |
| **HMVAE (this work)** | 6 | 147.9 ± 3.3 | 94.6 ± 1.0 | 1.8 ± 0.3 | 3.6 ± 0.6 | 0.26 ± 0.0 | 48.9 ± 0.5 | 74.8 ± 3.0 | **59.1 ± 1.2** |

(b)

| Mean over | Effective Hierarchical Layers | $p(\boldsymbol{x}_1)$ vs. $p(\boldsymbol{x}_1\|\boldsymbol{x}_2)$ | | | | | $p(\boldsymbol{x}_2)$ vs. $p(\boldsymbol{x}_2\|\boldsymbol{x}_1)$ | | |
|---|---|---|---|---|---|---|---|---|---|
| | | FID | Precision | Recall | F1 | Variance | Precision | Recall | F1 |
| - | 5 | 147.9 ± 3.3 | 94.6 ± 1.0 | 1.8 ± 0.3 | 3.6 ± 0.6 | 0.26 ± 0.0 | - | - | - |
| $\{z_{1,1}\}$ | 4 | 192.3 ± 0.4 | 83.7 ± 0.5 | 0.8 ± 0.6 | 1.6 ± 1.1 | 0.27 ± 0.0 | - | - | - |
| $\{z_{1,i}\}_{i \in \{1\}}$ | 3 | 273.0 ± 6.8 | 62.7 ± 2.0 | 0.0 ± 0.0 | 0.0 ± 0.0 | 0.29 ± 0.01 | - | - | - |
| $\{z_{1,i}\}_{i \in \{1,2\}}$ | 2 | 297.3 ± 1.6 | 43.3 ± 3.3 | 0.0 ± 0.0 | 0.0 ± 0.0 | 0.26 ± 0.01 | - | - | - |
| $\{z_{1,i}\}_{i \in \{1,2,3\}}$ | 1 | 300.3 ± 0.7 | 38.8 ± 0.7 | 0.0 ± 0.0 | 0.0 ± 0.0 | 0.13 ± 0.01 | - | - | - |
| - | 2 | - | - | - | - | - | 48.9 ± 0.5 | 74.8 ± 3.0 | 59.1 ± 1.2 |
| $\{z_{2,1}\}$ | 1 | - | - | - | - | - | 84.0 ± 2.1 | 7.2 ± 0.9 | 13.3 ± 1.5 |

"this flower has purple petals in the shape of a bell, with a star of yellow stamen in the center."

"flowers are alternately arranged,they are red in color"

MVAE  MMVAE  MDVAE  MHVAE  HMVAE (ablation)  HMVAE (proposed)

"this flower has purple petals in the shape of a bell, with a star of [...]"

"flowers are alternately arranged,they are red in color"

Mean from $z_{1,4}$   Mean from $z_{1,3}$   Mean from $z_{1,2}$   Mean from $z_{1,1}$

Figure 7: **Oxford Flower: generating images from captions.** Top: The penultimate column represents the HMVAE with two hierarchical levels for the images ($L(1) = 2$). The last column corresponds to the HMVAE with $L(1) = 5$, which is the best model considering all measures across all modalities. Bottom: as in the plot above, we generate $p(\boldsymbol{x}_i|\boldsymbol{g})$ from $q(\boldsymbol{g}|\boldsymbol{x}_{j\neq i})$. However, for the latent distributions starting at the latent variables indicated below the plot, we use the mean and not a conventional sample. We can thereby evaluate generated samples given varying degrees of hierarchical expressivity.

of experts posterior as the MMVAE, but adds hierarchical depth which correlates positively with variance and recall. Finally, we observe that the HMVAE improves the MHVAE across measures related to coherence (FID, image precision) and heterogeneity (FID, caption recall). Table 2b examines hierarchical decomposition in the HMVAE. We first sample conventionally from $q(\boldsymbol{g}|\boldsymbol{x}_{j\neq i})$. We then generate $p(\boldsymbol{x}_i|\boldsymbol{g})$ by using the conditional priors in the latent hierarchy for $\boldsymbol{x}_i$. Normally, we would sample conventionally from each of these latent distributions (rows 2 and 7). However, we

also evaluate models where we propagate the latent distribution means corresponding to the latent variables from the left column. In other words, we omit variations in some hierarchical levels. For the experiment on the image hierarchy, all measures improve with increased hierarchical expressivity. For the experiment on the caption hierarchy, greater hierarchical capacity is beneficial (F1) by increasing recall and decreasing precision. Increased recall is expected to correlate negatively with precision: generating a larger manifold typically increases the likelihood of generated samples beyond the target manifold.

**Qualitative results** Figure 7 focuses on $p(\boldsymbol{x}_1|\boldsymbol{x}_2)$. We generate $p(\boldsymbol{x}_1|\boldsymbol{g}^k)$ via $\boldsymbol{g}^{1:K} \sim q(\boldsymbol{g}|\boldsymbol{x}_2)$. For the MDVAE, we must additionally sample from the disentangled variable and hence generate images $p(\boldsymbol{x}_1|\boldsymbol{z}_1, \boldsymbol{g})$ from $\boldsymbol{z}_1 \sim p(\boldsymbol{z}_1)$ and $\boldsymbol{g} \sim q(\boldsymbol{g}|\boldsymbol{x}_2)$. Note that we always sample from the (conditional) priors over $\boldsymbol{z}$ across all multi-latent-variable models[3]. The MVAE's samples display a noticeable lack of coherence and fidelity which is in line with the results from Shi et al. (2019). The MM-VAE' samples have high coherence and low heterogeneity. The MDVAE's samples display precision but misalign some unimodal variations with the shared concept. The MHVAE's samples sometimes lack semantic coherence. The shallow HMVAE generates samples of good coherence and low heterogeneity. The deep HMVAE generates the most compelling samples relative to the baselines (as also indicated by the FID from Table 2a). Figure 7 presents the counterpart for the quantitative hierarchical ablation experiment from Table 2b. This figure demonstrates that the HMVAE decomposes image variation along its hierarchy.

## 5 CONCLUSION

We have proposed a hierarchical multimodal VAE (HMVAE) where unimodal latent hierarchies depend on a shared latent variable. We have demonstrated that the model improves generative modeling performance on multimodal data. Future work may further improve sample quality, for example, by incorporating different ways to parameterize the conditional densities: normalizing flows could improve the posterior's flexibility (Kingma et al., 2016), autoregressive decoders could explicitly represent the conditional dependencies across observed dimensions (Chen et al., 2018), and the transformer architecture may be a good candidate to replace the convolutional neural networks (Dosovitskiy et al., 2021). Note that we discuss ethical implications in App. E.

**Limitations** Our goal is the the representation of heterogeneity within modalities. Although the proposed HMVAE outperforms our baselines, the model is not perfect. For example, when generating images by sampling from the conditional distribution $p(\boldsymbol{x}_1|\boldsymbol{z}_1)$, where $\boldsymbol{z}_1 \sim p(\boldsymbol{z}_1|\boldsymbol{g})$ and $\boldsymbol{g} \sim q(\boldsymbol{g}|\boldsymbol{x}_2)$, the HMVAE must hallucinate a coherent set of visual representations (possibly conditional on a given caption). This can be challenging when the caption is vague and does not contain sufficient information for the inference network to infer the shared underlying structure correctly. In other words, the additional expressivity that the HMVAE obtains through the conditional latent variables also makes learning harder since $p(\boldsymbol{x}_1|\cdot)$ must represent a larger manifold for the proposed HMVAE than it does for the baselines.

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

## Appendix

We partition the Appendix into five sections:

A. **Additional experiments:** For the experiments from § 4.2, we provide further results on crossmodal generation. For the experiments from § 4.3, we provide results on unconditional generation.

B. **Likelihood estimation:** This section presents the estimators used to compute the likelihoods reported in Table 1.

C. **Implementation:** This section describes our implementation in detail. App. C.1 complements the background section § 2 by describing details of the employed hierarchical VAE architecture. This section also presents dataset-agnostic model specifications for the baselines. We then focus on dataset-specific implementation specifications, where App. C.2 covers feature generation on the CUB dataset, App. C.3 feature generation on the Oxford Flower dataset, and App. C.4 image generation on the Oxford Flower dataset.

D. **MDVAE: inference networks:** This section discusses an alternative inference network choice for multimodal disentanglement VAEs (MDVAEs).

E. **Ethical statement:** This section discusses ethical implications of the proposed model, e.g., its societal impact.

# A  ADDITIONAL EXPERIMENTS

This section presents additional experimental results that complement § 4 from the main paper.

## A.1  FEATURE GENERATION

In this section, we maximize the likelihood of image features $x_1 \in \mathbb{R}^{2048}$ and caption features $x_2 \in \mathbb{R}^{1024}$ on the CUB dataset and the Oxford Flower dataset. We use the same experimental setup as in § 4.

**Image feature generation**  Figure 8a presents generated samples where the captions are from the training set. We observe that the MMVAE still lacks variety in its generations, which indicates that this tendency is not caused by poor generalization. Figure 8b presents generated samples where the captions are from the test set. The HMVAE can represent different backgrounds (e.g., various sea or forest settings), even though this information is missing in the conditioning captions.

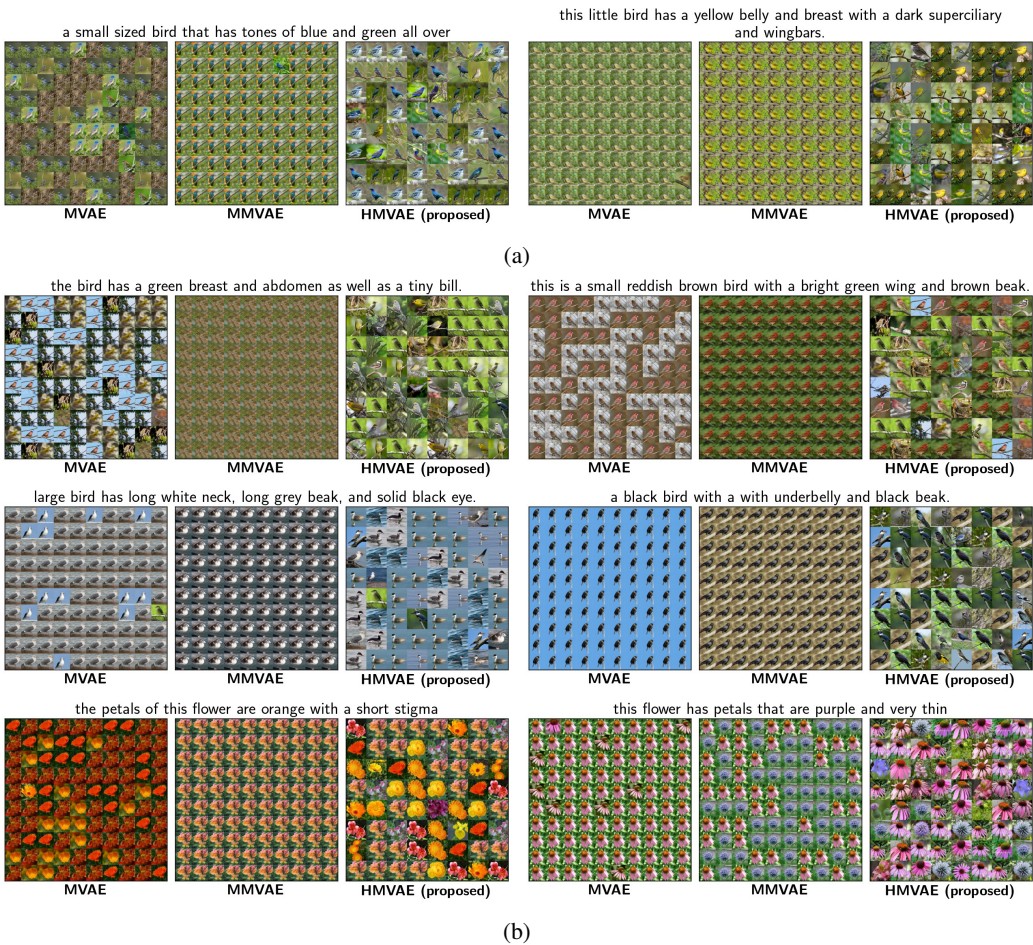

Figure 8: **CUB/Oxford Flower: Generating image feature vectors from caption feature vectors.** Plot (a) represents the training set, plot (b) the test set. We generate image features, look up the nearest-neighbor feature in the test set, and visualize the associated image.

**Caption feature generation**  Figure 9 presents additional results for caption feature generation which confirm our findings from § 4.

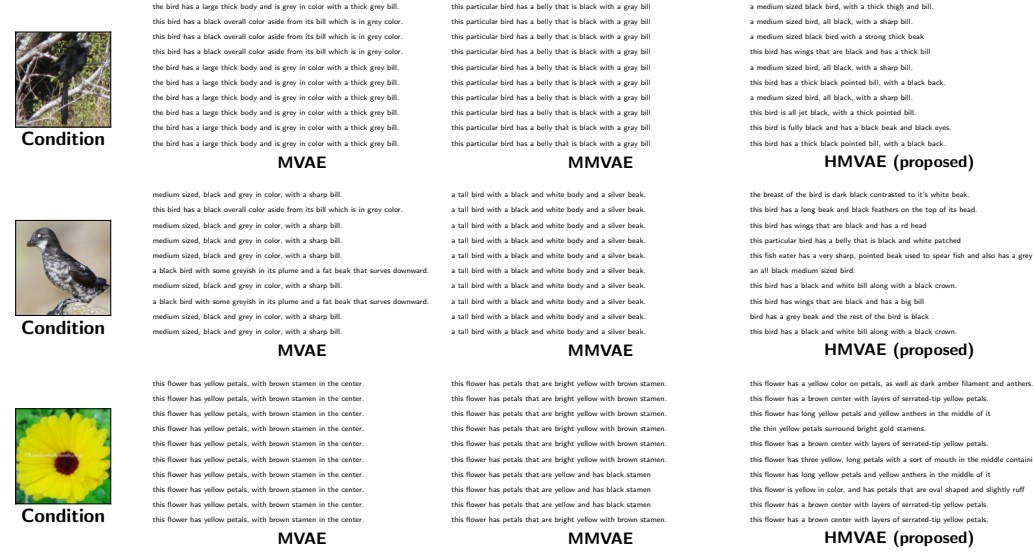

Figure 9: **CUB/Oxford Flower: Generating caption feature vectors from image feature vectors.** We generate caption features, look up the nearest-neighbor feature from the test set, and visualize the respective caption.

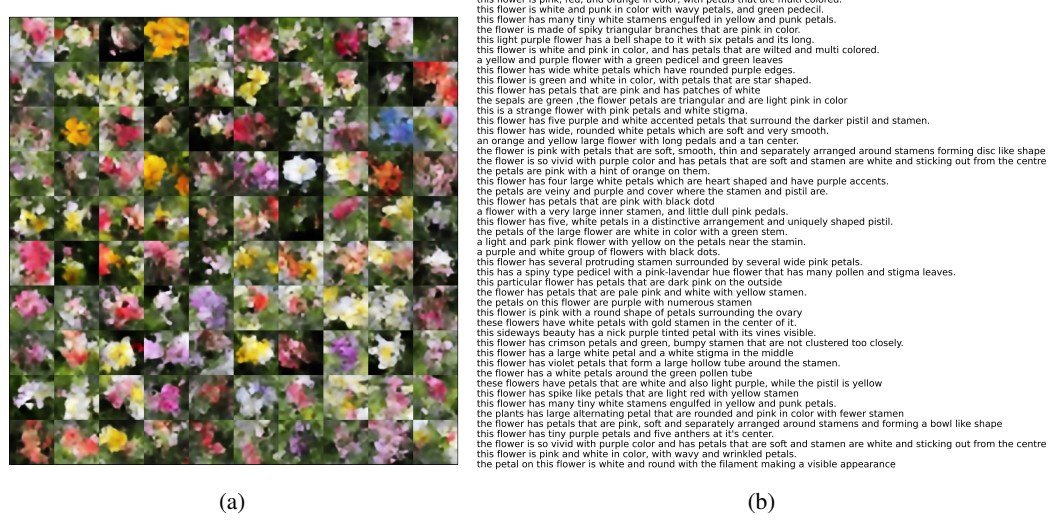

(a)                                                                        (b)

Figure 10: **Oxford Flower: Unconditional generation.** We generate $p(\boldsymbol{x}_i|\boldsymbol{g})$ from $\boldsymbol{g} \sim p(\boldsymbol{g})$ for $i \in \{1, 2\}$. In (b), we generate caption features, look up the nearest-neighbor feature from the test set, and visualize the respective caption.

## A.2 IMAGE GENERATION

In this section, we report additional results for § 4.3. Figure 10 visualizes unconditional samples. We observe that the HMVAE can generate diverse samples without relying on any conditioning.

# B  LIKELIHOOD ESTIMATION

In this section, we formalize the estimators that we use to evaluate the HMVAE, the MVAE (Wu and Goodman, 2018), and the MMVAE (Shi et al., 2019). We report all likelihoods in *nats*.

## B.1  JOINT LIKELIHOOD

We now focus on the joint likelihood over $M$ modalities. The special case with $M = 1$ describes the unimodal likelihoods $p(\boldsymbol{x}_i)$.

**Single latent variable**  We now discuss models that employ a single shared variable $\boldsymbol{g}$. We approximate the likelihood using $K$ importance samples:

$$\log p(\boldsymbol{x}_{1:M}) \approx \log \frac{1}{K} \sum_{k=1}^{K} \frac{p(\boldsymbol{x}_{1:M}, \boldsymbol{g}^k)}{q(\boldsymbol{g}^k | \boldsymbol{x}_{1:M})} \qquad \text{where} \quad \boldsymbol{g}^{1:K} \sim q(\boldsymbol{g} | \boldsymbol{x}_{1:M}) \qquad (7)$$

We follow Shi et al. (2019), evaluate the mixture of experts VAE using stratified sampling, and estimate a tighter estimate, which averages over modalities inside the logarithm. This objective should only be used during the evaluation because it may lead to an outsized impact of more informative modalities during optimization. Notably, $K$ samples of the joint posterior are equivalent to $T = K/M$ samples from each modality-specific posterior.

$$\log p(\boldsymbol{x}_{1:M}) \approx \log \frac{1}{M} \sum_{m=1}^{M} \frac{1}{T} \sum_{t=1}^{T} \frac{p(\boldsymbol{g}_m^t, \boldsymbol{x}_{1:M})}{q(\boldsymbol{g}_m^t | \boldsymbol{x}_m)} \qquad \text{where} \quad \boldsymbol{g}_m^{1:T} \sim q(\boldsymbol{g} | \boldsymbol{x}_m) \qquad (8)$$

**Latent hierarchy**  We now discuss models that employ both a shared variable $\boldsymbol{g}$ and conditionally independent, modality-specific variables $\boldsymbol{z}_{m,i}$. We approximate the likelihood using $K$ importance samples and omit the super- and subscripts in the sampling definition for improved readability:

$$\log p(\boldsymbol{x}_{1:M}) \approx \log \frac{1}{K} \sum_{k=1}^{K} \frac{p(\boldsymbol{x}_{1:M}, \boldsymbol{g}^k, \boldsymbol{z})}{q(\boldsymbol{g}^k, \boldsymbol{z} | \boldsymbol{x}_{1:M})} \qquad (9)$$
$$\text{where} \quad \boldsymbol{g}, \boldsymbol{z} \sim q(\boldsymbol{g}, \boldsymbol{z} | \boldsymbol{x}_{1:M})$$

The factorized generative and inference networks for the proposed latent hierarchy of the HMVAE are formalized in Eqs. 4 and 5 of the main paper. Note that both product and mixture of experts formulations of the shared posterior can be used within these hierarchical formulations.

## B.2  CROSSMODAL LIKELIHOOD

In this section, we discuss estimators for the crossmodal likelihood $p(\boldsymbol{x}_t | \boldsymbol{x}_c)$ given a target modality $t$ and a conditioning modality $c$. Analogously to Eq. 7, we evaluate the likelihoods using $K$ importance samples.

**Single latent variable**

We first formalize the conditional likelihood similarly to Suzuki et al. (2016) and Wu and Goodman (2018):

$$\log p(\boldsymbol{x}_t | \boldsymbol{x}_c) = \log \int_{\boldsymbol{g}} p(\boldsymbol{x}_t, \boldsymbol{g} | \boldsymbol{x}_c) d\boldsymbol{g} = \log \int_{\boldsymbol{g}} \frac{p(\boldsymbol{x}_t, \boldsymbol{x}_c, \boldsymbol{g})}{p(\boldsymbol{x}_c)} d\boldsymbol{g}$$
$$= \underbrace{\log \mathbb{E}_{q(\boldsymbol{g} | \boldsymbol{x}_c)} \left[ \frac{p(\boldsymbol{x}_t, \boldsymbol{x}_c, \boldsymbol{g})}{q(\boldsymbol{g} | \boldsymbol{x}_c)} \right]}_{①} - \underbrace{\log \mathbb{E}_{q(\boldsymbol{g} | \boldsymbol{x}_c)} \left[ \frac{p(\boldsymbol{x}_c, \boldsymbol{g})}{q(\boldsymbol{g} | \boldsymbol{x}_c)} \right]}_{②} \qquad (10)$$

In Eq. 10, term ① constitutes an estimation of the joint likelihood $p(\boldsymbol{x}_{1:M})$ using an importance distribution that is conditioned on modality $c$. Term ② estimates the unimodal likelihood $p(\boldsymbol{x}_c)$.

**Latent hierarchy** We now generalize the estimator from Eq. 10 to the proposed latent hierarchy of the HMVAE. We first formalize the conditional likelihood:

$$p(\boldsymbol{x}_t|\boldsymbol{x}_c) = \int_{\boldsymbol{g},\boldsymbol{z}} p(\boldsymbol{x}_t,\boldsymbol{g},\boldsymbol{z}|\boldsymbol{x}_c)d\boldsymbol{z}d\boldsymbol{g} = \int_{\boldsymbol{g},\boldsymbol{z}} \frac{p(\boldsymbol{x}_t,\boldsymbol{g},\boldsymbol{z},\boldsymbol{x}_c)}{p(\boldsymbol{x}_c)}d\boldsymbol{z}d\boldsymbol{g} \tag{11}$$

We assume that the modality-specific generative networks are conditionally independent given the shared representation (§ 3):

$$p(\boldsymbol{x}_t|\boldsymbol{x}_c) = \int_{\boldsymbol{g},\boldsymbol{z}} \frac{p(\boldsymbol{x}_t,\boldsymbol{z}_t|\boldsymbol{g})p(\boldsymbol{x}_c,\boldsymbol{z}_c|\boldsymbol{g})p(\boldsymbol{g})}{p(\boldsymbol{x}_c)}d\boldsymbol{z}d\boldsymbol{g} \tag{12}$$

We then define the importance distributions as the posterior over $\boldsymbol{g}$ and the conditional prior over $\boldsymbol{z}$. The method must learn $p(\boldsymbol{x}_t|\boldsymbol{z}_t)$ with $\boldsymbol{z}_t$ being solely conditioned on the *other* modality $c$. This formulation allows a fair comparison to the non-hierarchical baseline, which must do the same. In contrast, an importance distribution $q(\boldsymbol{z}_t|\boldsymbol{x}_t,\boldsymbol{g})$ would make the conditioning of $p(\boldsymbol{x}_t|\boldsymbol{z}_t)$ multimodal.

$$\log p(\boldsymbol{x}_t|\boldsymbol{x}_c) = \log \mathbb{E}_{q(\boldsymbol{g},\boldsymbol{z}_c|\boldsymbol{x}_c),p(\boldsymbol{z}_t|\boldsymbol{g})}\left[\underbrace{p(\boldsymbol{x}_t|\boldsymbol{z}_t,\boldsymbol{g})}_{①}\frac{p(\boldsymbol{x}_c,\boldsymbol{g},\boldsymbol{z}_c)}{q(\boldsymbol{g},\boldsymbol{z}_c|\boldsymbol{x}_c)}\right] - \log \mathbb{E}_{q(\boldsymbol{g},\boldsymbol{z}_c|\boldsymbol{x}_c)}\left[\frac{p(\boldsymbol{x}_c,\boldsymbol{g},\boldsymbol{z}_c)}{q(\boldsymbol{g},\boldsymbol{z}_c|\boldsymbol{x}_c)}\right]$$
$$\tag{13}$$

In Eq. 13, we can amortize computation for the unimodal likelihood $p(\boldsymbol{x}_c)$ which was defined in App. B.1. Term ① is the only one that requires additional forward passes through the network. Note that the remaining terms do not depend on the adjusted importance distribution $p(\boldsymbol{z}_t|\boldsymbol{g})$.

## C   IMPLEMENTATION

### C.1   GENERAL SPECIFICATIONS

**Hierarchical models**   The following formalizes latent distributions assuming the unimodal hierarchical VAE inspired by Sønderby et al. (2016) from § 2. We start with the conditional isotropic Gaussian prior

$$
\begin{aligned}
&p_{\phi,\theta}(\boldsymbol{z}_i|\boldsymbol{z}_{i+1}) \\
&= \mathcal{N}(\boldsymbol{z}_i|\mu_\theta(d_{\phi,\theta,i}(\boldsymbol{z}_{i+1})), \sigma_\theta(d_{\phi,\theta,i}(\boldsymbol{z}_{i+1}))),
\end{aligned}
\tag{14}
$$

where $i \in \{1, ..., L\}$ refers to the hierarchical level. The MLP $d_{\phi,\theta,i}$ represents the top-down network, which is partially shared across the inference and generative networks. Its output is passed to the stochastic layers $\mu_\theta$ and $\sigma_\theta$ that parameterize the prior distribution.

We then focus on the more complicated case of inferring the isotropic Gaussian posterior $q_{\phi,\theta}(\boldsymbol{z}_i|\boldsymbol{z}_{i+1}, \boldsymbol{x})$ where $i < L$ for $L$ latent variables (Fig. 11). We first infer the hidden states for $\boldsymbol{z}_{i+1}$, $\boldsymbol{x}$ and for hierarchical level $i \in \{1, ..., L\}$, where $d_{\theta,\phi}$ is the top-down MLP and $d_\phi$ the bottom-up MLP:

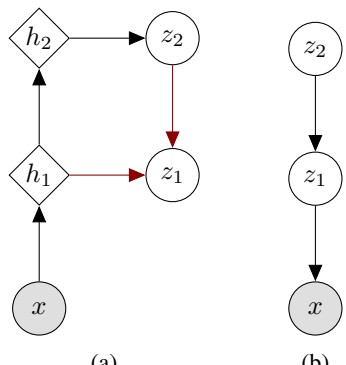

(a)                    (b)

Figure 11:   **Unimodal hierarchical VAEs** (Sønderby et al., 2016) (a) Inference network. The red edges visualize the merging procedure of bottom-up and top-down information. (b) Generative network.

$$
\begin{aligned}
\boldsymbol{h}_{i,t} &= d_{\phi,\theta,i}(\boldsymbol{z}_{i+1}), \\
\boldsymbol{h}_{i,b} &= d_{\phi,i}(\boldsymbol{x}).
\end{aligned}
\tag{15}
$$

We concatenate both hidden states along the first dimension and pass the result through another neural network $m_\phi$ to create the joint representation. We denote the concatenation operation as $\langle \cdot, \cdot \rangle$:

$$
\boldsymbol{h}_{i,j} = m_\phi(\langle \boldsymbol{h}_{i,t}, \boldsymbol{h}_{i,b} \rangle).
\tag{16}
$$

This joint representation is input to the stochastic layers $\mu_\phi$ and $\sigma_\phi$ that parameterize the posterior distribution:

$$
q_{\phi,\theta}(\boldsymbol{z}_i|\boldsymbol{z}_{i+1}, \boldsymbol{x}) = \mathcal{N}(\boldsymbol{z}_i|\mu_\phi(\boldsymbol{h}_{i,j}), \sigma_\phi(\boldsymbol{h}_{i,j})).
\tag{17}
$$

The remaining priors and posteriors are parameterized similarly. *Appendix A* from Maaløe et al. (2019) provides further background on unimodal hierarchical VAEs that inspire this work.

**MVAE and MMVAE**   We follow Shi et al. (2019), *train* the MMVAE by maximizing a loose bound and *evaluate* using a tight estimate as described in App. B.1. The following describes exemplary differences in implementation compared to the original formulations:

- We employ a slightly different network architecture for the MVAE and the MMVAE (Wu and Goodman, 2018) relative to their original formulation, respectively. For example, we do not include dropout regularization across all implementations.

- For the MMVAE, we neither employ importance weighted autoencoders (Burda et al., 2015) nor use a doubly reparameterized gradient estimator (Tucker et al., 2018). Note that these techniques increase computational requirements. Neither the MVAE nor the HMVAE relies on such measures.

- We do not use Laplace latent distributions in the MMVAE, but Gaussian latent distributions as in all implemented methods.

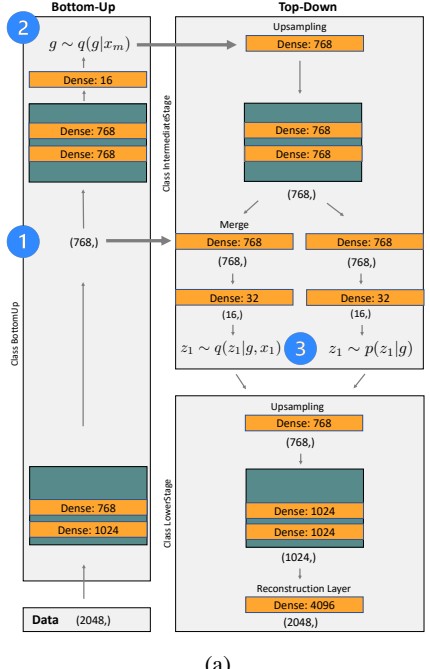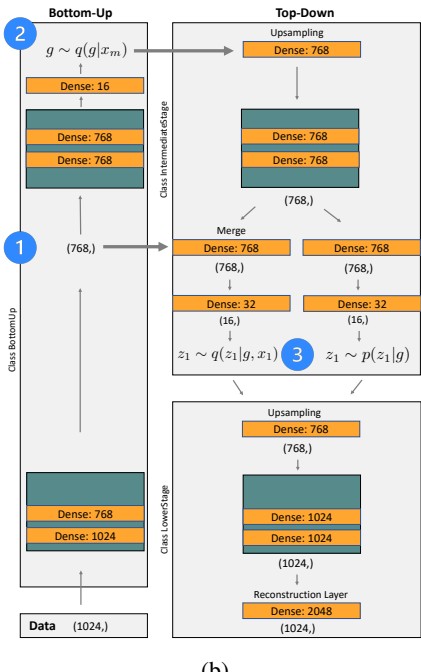

|  | |
|---|---|
| (a) | (b) |

Figure 12: **Feature genereation on CUB/Oxford Flower – HMVAE.** (a) Image ft. VAE. (b) Caption ft. VAE. Both decoders parameterizes normal distributions. We do not display the *LeakyReLU* activation functions.

## C.2 FEATURE GENERATION: CUB DATASET

**Data** The CUB dataset includes an image and a caption modality. For the images, we use the features vectors $x_1 \in \mathbb{R}^{2048}$ provided by Xian et al. (2018a), who employed a ResNet-101 (He et al., 2016) trained on ImageNet (Deng et al., 2009). As described in § 4, we use caption features $x_2 \in \mathbb{R}^{1024}$. We divide the dataset into three splits that do not share classes. The training set consists of 111 classes. The validation split consists of 39 classes and the test split contains 50 classes. We train the model on the training set and tune hyperparameters on the validation set. We then evaluate the trained model on the test set, i.e., we do not use the validation set for training as in some previous works. We use the same test split as Reed et al. (2016) who provided the CNN-RNN feature extractor for the captions, which is essential to avoid test data leakage. We standardize the features for both modalities, i.e., we subtract the mean $\mu$ and divide by the standard deviation $\sigma$. We compute both parameters on the training set. We assume that such preprocessing can be helpful, because the models generate a Gaussian likelihood $p(x_i|\cdot)$.

**All models** The generative models maximize the likelihood of the data under learned Gaussian distributions. We train the methods for 100 epochs with a learning rate of 1e-4 and a batch size of 256. For each datapoint in a batch, we use ten samples from the respective shared posterior (i.e., $g^{1:10} \sim q(g|x_{1:M})$) during training. We gradually increase the KL-regularization weighting factor from zero to one for a warm-up period of 25 (Sønderby et al., 2016).

**Proposed HMVAE** The hierarchical VAEs are visualized in Fig. 12a and Fig. 12b. The shared latent space $g$ contains 16 dimensions.

**MVAE and MMVAE** The models are visualized in Table 3 and Table 3. We share the architecture across baselines. Those models are similar in their architecture to the hierarchical method, but lack the hierarchical components. We tested a version with additional layers to compensate for this reduced capacity. However, models with more capacity (i.e., more weights) did not improve the performance of the flat models. The shared latent space $g$ contains eight dimensions.

Table 3: **Feature generation on CUB/Oxford Flower – MVAE and MMVAE.** Both decoders parameterize normal distributions.

| Image ft. encoder | Image ft. decoder | Caption ft. encoder | Caption ft. decoder |
|---|---|---|---|
| Input $\in \mathbb{R}^{2048}$ | Input $\in \mathbb{R}^{16}$ | Input $\in \mathbb{R}^{1024}$ | Input $\in \mathbb{R}^{16}$ |
| Dense(2048, 1024) | Dense(16, 768) | Dense(1024, 1024) | Dense(16, 768) |
| LeakyReLU | LeakyReLU | LeakyReLU | LeakyReLU |
| Dense(1024, 768) | Dense(768, 768) | Dense(1024, 768) | Dense(768, 768) |
| LeakyReLU | LeakyReLU | LeakyReLU | LeakyReLU |
| Dense(768, 768) | Dense(768, 1024) | Dense(768, 768) | Dense(768, 1024) |
| LeakyReLU | LeakyReLU | LeakyReLU | LeakyReLU |
| Dense(768, 768) | Dense(1024, 1024) | Dense(768, 768) | Dense(1024, 1024) |
| LeakyReLU | LeakyReLU | LeakyReLU | LeakyReLU |
| Dense(768, 32) | Dense(1024, 4096) | Dense(768, 32) | Dense(1024, 2048) |

## C.3 FEATURE GENERATION: OXFORD FLOWER DATASET

**Data** The Oxford Flower dataset includes an image and a caption modality. We use feature vector representations of the images, where $x_1 \in \mathbb{R}^{2048}$, and of the captions, where $x_2 \in \mathbb{R}^{1024}$. We standardize the features for both modalities, i.e., we subtract the mean $\mu$ and divide by the standard deviation $\sigma$. We compute both parameters on the training set. We assume that such preprocessing can be helpful, because the respective models generate a Gaussian likelihood $p(x_i|\cdot)$. We divide the dataset into three splits that do not share classes. The training set consists of 62 classes. Both validation and test split contain 20 classes. We train the model on the training set and tune hyperparameters on the validation set. We then evaluate the trained model on the test set, i.e., we do not use the validation set for training as in some previous works. We use the same test split as Reed et al. (2016) who provided the CNN-RNN feature extractor for the captions, which is essential to avoid test data leakage.

**All models** The VAEs maximize the likelihood of each modality under a learned Gaussian distribution. All models are trained for 100 epochs. We gradually increase the KL-regularization weighting factor from zero to one for a warm-up period of 25 (Sønderby et al., 2016). We train the models with a learning rate of 1e-4 and a batch size of 256. For each datapoint in a batch, we use ten samples from the respective shared posterior (i.e., $g^{1:10} \sim q(g|x_{1:M})$) during training.

**Proposed HMVAE** Figure 12a displays the image VAE, Fig. 12b shows the caption feature vector VAE.

**MVAE and MMVAE** Table 3 displays the image VAE, Table 3 shows the caption feature vector VAE.

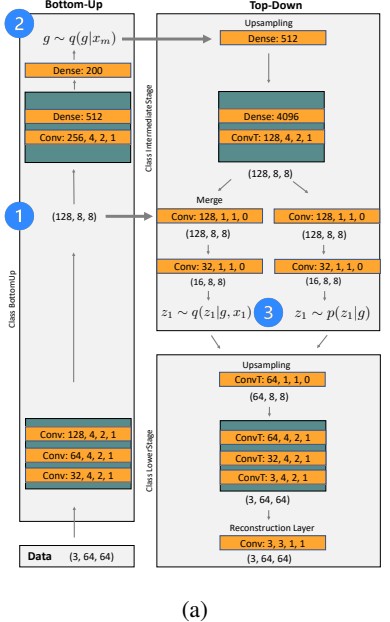 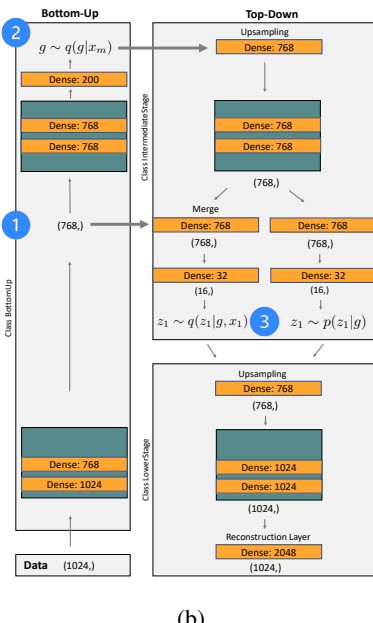

(a)                                                                      (b)

Figure 13: **Image generation on Oxford Flower – HMVAE.** (a) Image VAE. The decoder generates scalars, which are used to compute the binary cross-entropy with the ground truth. We do not display the *Gelu* activation functions (Hendrycks and Gimpel, 2016). (b) Caption ft. VAE. The decoder parameterizes a normal distribution. We do not display the *LeakyReLU* activation functions.

Table 4: **Image generation on Oxford Flower – number of trainable parameters.** The second column represents the number of latent variables. One of the reason the deeper HMVAE has fewer parameters than the shallow HMVAE lies in the dense layer that maps between spatial (unimodal) and vector (shared) representations. The deeper HMVAE achieves a lower spatial resolution and which minimizes the parameters of this layer. Note that this single layer can easily reach more than three million parameters, see the official MVAE implementation at `www.github.com/mhw32/multimodal-vae-public/blob/master/celeba/model.py`. In general, we want to primarily ensure that the HMVAE does not improve performance due to excessive capacity.

| Model Name | # LV | # Params |
|---|---|---|
| MVAE (Wu and Goodman, 2018) | 1 | 19.2M |
| MMVAE (Shi et al., 2019) | 1 | 19.2M |
| MDVAE (Mahajan et al., 2020), i.a. | 3 | 13.3M |
| MHVAE (Vasco et al., 2020) | 3 | 16.8M |
| HMVAE (shallow) | 3 | 15.9M |
| **HMVAE (this work)** | 6 | 11.6M |

### C.4 IMAGE GENERATION: OXFORD FLOWER DATASET

**Data** We preprocess the images by first making them quadratic (using *CenterCrop* in PyTorch) and then resizing them to $x_1 \in \mathbb{R}^{3 \times 64 \times 64}$. Note that we described general specifications in App. C.3.

**All models** We train the models with a learning rate of 1e-4 and a batch size of 64.

**Proposed HMVAE (shallow hierarchy)** Figure 13a portrays the image VAE. Figure 13b presents the hierarchical VAE for the other modality. We gradually increase the KL-regularization weighting factor from zero to one for a warm-up period of 20 (Sønderby et al., 2016).

**Proposed HMVAE (deep hierarchy)** To insert further hierarchical levels, we *"slice"* deterministic blocks. We then add stochastic layers and respective pre- and postprocessing layers for each hierar-

chical level. Note that these layers often add just a few trainable parameters because of 1x1 convolutions or small channel sizes for the stochastic variables. Note that the deeper model uses convolutions blocks of three layers instead of single convolutional layers. Input and output channel dimension are identical on the outside and reduced within the block. We fix the channel size across the deterministic parts of the model and gradually down- or upsample the spatial dimension. We downsample with average pooling and upsample using nearest-neighbor interpolation (Child, 2021). We fix the channels for the latent variables along the image hierarchy to 16 and solely adjust the spatial dimensions. We do not use any warm-up scheme for the KL-divergence term in the loss, i.e., we use the natural ELBO.

**MVAE and MMVAE** The MVAE (Wu and Goodman, 2018) and MMVAE (Shi et al., 2019) share their basic architecture. Table 5 displays the image VAE, Table 5 shows the caption feature VAE. We add extra capacity to the decoder to compensate for the missing parameters from hierarchical components. We gradually increase the KL-regularization weighting factor from zero to one for a warm-up period of 20 (Sønderby et al., 2016).

Table 5: **Image generation on Oxford Flower – MVAE and MMVAE.** The image decoder generates scalars, which are used to compute the binary cross-entropy with the ground truth. The caption decoder parameterizes a normal distribution. We use the *Swish* activation function (Ramachandran et al., 2017).

| Image ft. encoder | Image ft. decoder | Caption ft. encoder | Caption ft. decoder |
|---|---|---|---|
| Input $\in \mathbb{R}^{3 \times 64 \times 64}$ | Input $\in \mathbb{R}^{100}$ | Input $\in \mathbb{R}^{1024}$ | Input $\in \mathbb{R}^{16}$ |
| Conv(3, 32, 4, 2, 1) | Dense(100, 512) | Dense(1024, 1024) | Dense(16, 768) |
| Swish | Swish | LeakyReLU | LeakyReLU |
| Conv(32, 64, 4, 2, 1) | Dense(512, 4096) | Dense(1024, 768) | Dense(768, 768) |
| Swish | Swish | LeakyReLU | LeakyReLU |
| Conv(64, 128, 4, 2, 1) | ConvT(256, 128, 4, 2, 1, 0) | Dense(768, 1024) | Dense(768, 768) |
| Swish | Swish | LeakyReLU | LeakyReLU |
| Conv(64, 128, 1, 1, 0) | ConvT(128, 64, 1, 1, 0, 0) | Dense(1024, 768) | Dense(1024, 1024) |
| Swish | Swish | LeakyReLU | LeakyReLU |
| Conv(128, 256, 4, 2, 1) | ConvT(64, 64, 1, 1, 0, 0) | Dense(768, 768) | Dense(1024, 1024) |
| Swish | Swish | LeakyReLU | LeakyReLU |
| Dense(4096, 512) | ConvT(64, 64, 4, 2, 1, 0) | Dense(768, 768) | Dense(1024, 1024) |
| Swish | Swish | LeakyReLU | LeakyReLU |
| Dense(512, 200) | ConvT(64, 32, 4, 2, 0, 0) | Dense(768, 32) | Dense(1024, 1024) |
| | Swish | | LeakyReLU |
| | ConvT(32, 3, 4, 2, 0, 0) | | Dense (1024, 1024) |
| | Swish | | LeakyReLU |
| | Conv(3, 3, 4, 2, 1) | | Dense(1024, 1024) |
| | Sigmoid | | LeakyReLU |
| | | | Dense(1024, 2048) |

**MDVAE** There are many implementation choices for MDVAEs (Huang et al., 2018; Hsu and Glass, 2018; Mahajan et al., 2020; Sutter et al., 2020; Daunhawer et al., 2021b; Lee and Pavlovic, 2021). However, many related works employ a product of experts posterior (Eq. 2). This posterior choice can produce poor results given the type of data considered in this work (where there is significant variation in both modalities) (Shi et al., 2019). Therefore, we use a mixture of experts posterior (Eq. 3) as in the HMVAE. The implemented MDVAE is almost identical to the HMVAE– except that the unimodal variables $z$ are marginally independent from $g$ (Figs. 2b and 2c). We train the model by maximizing the ELBO:

$$\log p_\theta(\boldsymbol{x}_{1:M}) \geq \mathbb{E}_{q_\phi}\left[\log \frac{p_\theta(\boldsymbol{g})\prod_{i=1}^{M} p_\theta(\boldsymbol{z}_i)}{q_\phi(\boldsymbol{g}|\boldsymbol{x}_{1:M})\prod_{i=1}^{M} q_\phi(\boldsymbol{z}_i|\boldsymbol{x}_i)} + \log \prod_{i=1}^{M} p_\theta(\boldsymbol{x}_i|\boldsymbol{g}, \boldsymbol{z}_i)\right], \quad (18)$$

where $\boldsymbol{g} \sim q_\phi(\boldsymbol{g}|\boldsymbol{x}_{1:M})$ and $\boldsymbol{z}_i \sim q_\phi(\boldsymbol{z}_i|\boldsymbol{x}_i)$ with $i \in \{1, ..., M\}$. We use $\boldsymbol{z}_1 \in \mathbb{R}^8, \boldsymbol{g} \in \mathbb{R}^{100}, \boldsymbol{z}_2 \in \mathbb{R}^8$ and find that increasing the unimodal latent sizes tends to decrease crossmodal coherence. We gradually increase the KL-regularization weighting factor from zero to one for a warm-up period of 20 (Sønderby et al., 2016). Table 6 shows the implementation.

**MHVAE** We follow Vasco et al. (2020) (the MHVAE authors) and implement the generative network from Fig. 3a for two hierarchical levels and the inference network from Fig. 2c. As done by

Table 6: **Image generation on Oxford Flower – MDVAE.** For both models, the encoder produces a single tensor of size $D$. We split this tensor into two parts that represent $\boldsymbol{g} \in \mathbb{R}^{100}$ and $\boldsymbol{z}_i \in \mathbb{R}^8$ for $i \in \{1, 2\}$, respectively. The decoder for $\boldsymbol{x}_i$ uses a concatenation of $\boldsymbol{z}_i$ and $\boldsymbol{g}$ as input, where again $i \in \{1, 2\}$. For the image VAE, the decoder generates scalars, which are used to compute the binary cross-entropy with the ground truth. For the caption VAE, the decoder parameterize normal distributions.

| Image encoder | Image decoder | Caption encoder | Caption decoder |
|---|---|---|---|
| Input $\in \mathbb{R}^{3\times64\times64}$ | Input $\in \mathbb{R}^{100}$ | Input $\in \mathbb{R}^{1024}$ | Input $\in \mathbb{R}^{100}$ |
| Conv(3, 32, 4, 2, 1) | Dense(100, 256 * 5 * 5) | Dense(1024, 1024) | Dense(16, 768) |
| Swish | Swish | LeakyReLU | LeakyReLU |
| Conv(32, 64, 4, 2, 1) | ConvT(256, 128, 4, 1, 0) | Dense(1024, 768) | Dense(768, 768) |
| BatchNorm | BatchNorm | LeakyReLU | LeakyReLU |
| Swish | Swish | Dense(768, 768) | Dense(768, 1024) |
| Conv(64, 128, 4, 2, 1) | ConvT(128, 64, 4, 2, 1) | LeakyReLU | LeakyReLU |
| BatchNorm | BatchNorm | Dense(768, 768) | Dense(1024, 1024) |
| Swish | Swish | LeakyReLU | LeakyReLU |
| Conv(128, 256, 4, 1, 0) | ConvT(64, 32, 4, 2, 1) | Dense(768, 200) | Dense(1024, 2048) |
| BatchNorm | BatchNorm | | |
| Swish | Swish | | |
| Dense(256*5*5, 512) | ConvT(32, 3, 4, 2, 1) | | |
| LeakyReLU | | | |
| Dense(512, 200) | | | |

the authors, we employ *domain dropout* to learn representations over multiple modalities. We use a uniform domain dropout distribution, i.e., maximize three ELBOs given importance samples from $q(\boldsymbol{g}|\boldsymbol{x}_1)$, $q(\boldsymbol{g}|\boldsymbol{x}_2)$, and $q(\boldsymbol{g}|\boldsymbol{x}_{1:M})$, respectively (Eq. 19, similar to Wu and Goodman (2018)).

We follow Vasco et al. (2020) and maximize the sum over the multi- and unimodal ELBOs:

$$\mathcal{L} = \mathcal{L}_{\text{joint}} + \mathcal{L}_{\text{images}} + \mathcal{L}_{\text{captions}}$$

$$\mathcal{L}_{\text{joint}} := \sum_{i=1}^{M} \mathbb{E}_{q_\phi(\boldsymbol{z}_i|\boldsymbol{x}_i)}\left[\log p_\theta(\boldsymbol{x}_i|\boldsymbol{z}_i)\right] - \sum_{i=1}^{M} \mathbb{E}_{q_\phi(\boldsymbol{z}_i|\boldsymbol{x}_i)}\left[\log \frac{p_\theta(\boldsymbol{z}_i|\boldsymbol{g})}{q_\phi(\boldsymbol{z}_i|\boldsymbol{x}_i)}\right]$$
$$- \mathbb{E}_{q_\phi(\boldsymbol{g}|\boldsymbol{x}_{1:M})}\left[\log \frac{p_\theta(\boldsymbol{g})}{q_\phi(\boldsymbol{g}|\boldsymbol{x}_{1:M})}\right]$$

$$\mathcal{L}_{\text{images}} := \sum_{i=1}^{M} \mathbb{E}_{q_\phi(\boldsymbol{z}_i|\boldsymbol{x}_i)}\left[\log p_\theta(\boldsymbol{x}_i|\boldsymbol{z}_i)\right] - \sum_{i=1}^{M} \mathbb{E}_{q_\phi(\boldsymbol{z}_i|\boldsymbol{x}_i)}\left[\log \frac{p_\theta(\boldsymbol{z}_i|\boldsymbol{g})}{q_\phi(\boldsymbol{z}_i|\boldsymbol{x}_i)}\right] \quad (19)$$
$$- \mathbb{E}_{q_\phi(\boldsymbol{g}|\boldsymbol{x}_1)}\left[\log \frac{p_\theta(\boldsymbol{g})}{q_\phi(\boldsymbol{g}|\boldsymbol{x}_1)}\right]$$

$$\mathcal{L}_{\text{captions}} := \sum_{i=1}^{M} \mathbb{E}_{q_\phi(\boldsymbol{z}_i|\boldsymbol{x}_i)}\left[\log p_\theta(\boldsymbol{x}_i|\boldsymbol{z}_i)\right] - \sum_{i=1}^{M} \mathbb{E}_{q_\phi(\boldsymbol{z}_i|\boldsymbol{x}_i)}\left[\log \frac{p_\theta(\boldsymbol{z}_i|\boldsymbol{g})}{q_\phi(\boldsymbol{z}_i|\boldsymbol{x}_i)}\right]$$
$$- \mathbb{E}_{q_\phi(\boldsymbol{g}|\boldsymbol{x}_2)}\left[\log \frac{p_\theta(\boldsymbol{g})}{q_\phi(\boldsymbol{g}|\boldsymbol{x}_2)}\right]$$

We gradually increase a factor before the KL-divergence term from zero to one for five epochs (unimodal variables) or ten epochs (shared variable) as done by Vasco et al. (2020). We use $\boldsymbol{z}_1 \in \mathbb{R}^{256}, \boldsymbol{g} \in \mathbb{R}^{100}, \boldsymbol{z}_2 \in \mathbb{R}^8$. Tables 7 and 8 present the architecture.

Table 7: **Image generation on Oxford Flower – MHVAE (1)**
Networks for $x_1$: We follow Vasco et al. (2020) and use the miniature DCGAN architecture (Radford et al., 2015). The decoder generates scalars, which are used to compute the binary cross-entropy with the ground truth. Networks for $x_2$: The decoder for the second modality parameterizes a normal distribution.

| Lower encoder: $x_1 \to h_1$ | Lower decoder: $z_1 \to x_1$ | Lower encoder: $x_2 \to h_2$ | Lower decoder: $z_2 \to x_2$ |
|---|---|---|---|
| Input $\in \mathbb{R}^{3 \times 64 \times 64}$ | Input $\in \mathbb{R}^{128}$ | Input $\in \mathbb{R}^{1024}$ | Input $\in \mathbb{R}^{48}$ |
| Conv(3,32,4,2,1) | Dense(128, 256*5*5) | Dense(1024, 1024) | Dense(48, 768) |
| Swish | ConvT(256, 128, 4, 1, 0) | LeakyReLU | LeakyReLU |
| Conv(32, 64, 4, 2, 1) | BatchNorm | Dense(1024, 768) | Dense(768, 1024) |
| Swish | Swish | LeakyReLU | LeakyReLU |
| Conv(64, 128, 4, 2, 1) | ConvT(128, 64, 4, 2, 1) | Dense(768, 768) | Dense(1024, 1024) |
| BatchNorm | BatchNorm | LeakyReLU | LeakyReLU |
| Swish | Swish | Dense(768, 512) | Dense(1024, 1024*2) |
| Conv(128, 256, 4, 1, 0) | ConvT(64, 32, 4, 2, 1) | LeakyReLU | |
| BatchNorm | BatchNorm | | |
| Swish | Swish | | |
| Dense(256*5*5, 512) | ConvT(32, 3, 4, 2, 1) | | |
| Swish | | | |

Table 8: **Image generation on Oxford Flower – MHVAE (2)**
The networks for $g \to z$ and $h \to z$ are identical across modalities – except for the final output dimension which depends on the latent size, where $z_1 \in \mathbb{R}^{128}$ and $z_2 \in \mathbb{R}^{48}$.

| Upper encoder: $h \to g$ | Upper decoder: $g \to z$ | Encoder: $h \to z$ |
|---|---|---|
| Input $\in \mathbb{R}^{512*2}$ | Input $\in \mathbb{R}^{100}$ | Input $\in \mathbb{R}^{100}$ |
| Dense(512*2, 768) | Dense(-, 500) | Dense(512, -) |
| LeakyReLU | LeakyReLU | |
| Dense(768, 768) | Dense(500, 500) | |
| LeakyReLU | LeakyReLU | |
| Dense(768, 768) | Dense(500, 500) | |
| LeakyReLU | LeakyReLU | |
| Dense(768, 100*2) | Dense(500, -) | |

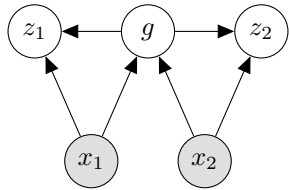

Figure 14: MDVAE: Alternative inference network

## D    MDVAE: ALTERNATIVE INFERENCE NETWORK

There are several inference network choices for multimodal disentanglement VAEs (MDVAEs): Fig. 2c depicts the natural inference network. Figure 14 modifies this network to include dependencies between $g$ and $z_{1:M}$. This modified architecture recovers our proposed hierarchical inference network for the special case of two hierarchical layers (Fig. 3b). Note that the edges between $g$ and $z_{1:2}$ are still absent in the generative model (Fig. 2b): the model must learn to represent *independent* variation in $z_1$. For example, the model must generate $p_\theta(x_i|z_i, g)$ from $z_i \sim p_\theta(z_1)$ (not necessarily $z_i \sim q_\phi(z_1|x_i, g)$) and $g \sim q_\phi(g|x_{j \neq i})$.

## E    ETHICS STATEMENT

Generative models can have pernicious effects on society via creating and propagating synthetic data that mimics reality (e.g., DeepFakes). Extending these models to multiple modalities can strengthen their performance and hence amplify these challenges. Therefore, annotating and marking data as synthetically generated is vital to ensure that people can spot and identify synthetic data outside of contexts where their synthetic nature is clear. Furthermore, the training data for deep generative models rarely represents the rich diversity of people, images, and objects in the world. Biases towards overrepresented groups will inevitably creep into generative models, and if the model is used for solving classification tasks on underrepresented groups, it is unlikely to prove useful. Extending the training data to multiple modalities can mitigate these effects if the additional modalities are less biased. In general, building generative models that can better cope with heterogeneity, e.g., by improving models or collecting more realistic data, is a good step towards alleviating harmful biases.

