# OpenReview forum: "Hierarchical Multimodal Variational Autoencoders"
_ICLR.cc/2022/Conference — ICLR 2022 Submitted_

### Official Review · Reviewer_Gac5 · 2021-10-20

**Correctness:** 3
**Technical Novelty And Significance:** 3
**Empirical Novelty And Significance:** 3
**Recommendation:** 6
**Confidence:** 3

**Main Review:**

Strengths
* a new approach incorporates a hierarchy between shared and modality-specific latent variables.
* Paper is generally well-written.

Weaknesses
* evaluation is restricted to dense modalities such as images and text.
* The paper can be improved if the author can provide further insight into why defining a hierarchy of latent variables provides a better result.
* The authors claim that a hierarchical latent representation is an inductive bias that captures realistic data variations and guides learning. It would be helpful if they demonstrate this by showing the representation of modality-specific variations with quantification.
* The authors mentioned that they found that increasing the unimodal latent sizes tends to decrease crossmodal coherence. It would be interesting to show this result.
* It would be helpful if the author can add a heterogeneity metric as their goal was identifying the representation of heterogeneity within modalities. The current evaluation metrics used in this paper are somewhat subjective and difficult to claim which method performs better.



**Summary Of The Paper:**

The authors proposed a hierarchical multimodal VAE (HMVAE) that represents modality-specific variations using latent variables dependent on a shared top-level variable. They demonstrate that the proposed approach can represent multimodal heterogeneity and outperform existing methods in sample generation quality and quantitative measures.

**Summary Of The Review:**

The authors proposed an HMVAE where unimodal latent variables depend on a shared latent variable.
They demonstrated that the model improves generative modeling performance on multimodal data but is restricted to dense modalities such as images and text.
The proposed method is well-motivated but the paper can be improved if the author clearly demonstrated how a hierarchy of latent variables improves identifing modality-specific latent variables.

---

> ### Author Response · Authors · 2021-11-23
> **Initial response to Reviewer Gac5**
>
> We thank Reviewer Gac5 for the feedback. We uploaded a revised version of the paper/code that incorporates this feedback. We highlight our changes in our response below.
>
> **“It would be helpful if the author can add a heterogeneity metric as their goal was identifying the representation of heterogeneity within modalities. The current evaluation metrics used in this paper are somewhat subjective and difficult to claim which method performs better.”**
>
> **“The authors claim that a hierarchical latent representation is an inductive bias that captures realistic data variations and guides learning. It would be helpful if they demonstrate this by showing the representation of modality-specific variations with quantification.”**
>
> We agree that a good multimodal VAE should capture both heterogeneity and modality-specific variations (coherence). To this end, the revised paper contains the following measures (some of which are new):
> Likelihood estimates (Tab. 1), which estimate that the approximate generative distribution matches the true distribution. Note that this measure was already present in the initial submission and measures both criteria.
> Frechet Inception Distances (Tab. 2) compare the means and covariance matrices of the true and approximate data and thereby also measure both criteria.
> The average variance of generated features (Tab. 2), which indicates heterogeneity.
> Precision (Tab. 2), which indicates how “modality-specific” the generated samples are.
> Recall (Tab. 2) which assesses heterogeneity.
>
> **“The paper can be improved if the author can provide further insight into why defining a hierarchy of latent variables provides a better result.”**
>
> We suggest that a hierarchical latent representation is an inductive bias that captures realistic data variations and guides learning. Such a hierarchy can capture dependencies between modality-exclusive and shared structure (see the revised Fig. 1). Deeper hierarchies can be beneficial as they allow factorizing variations across more levels of abstractions. We included further respective intuition in Section 3 and Fig. 3.
>
> Tab. 2 provides new results (Precision/Recall/F1, FID, sample variance) that substantiate improved performance. Fig. 7 displays a new qualitative comparison where our model improves semantic coherence. Tab. 2 now provides an ablation between the HMVAE with L(1)=1 (MMVAE), L(1)=2, and L(1)=5. We find that the hierarchical depth correlates positively with performance. Note that the MVAE attains the best FID and image F1 scores in Tab. 2. However, we find that the model’s qualitative samples lack fidelity (Fig. 7). Furthermore, the model scores exceptionally low on image precision and caption F1 scores.
>
> For the deep model, we also evaluate hierarchical decomposition. To this end, we perform an experiment where we first sample conventionally from q(g|x_i). We then sample either conventionally or the mean from any lower-level latent variable z. When sampling the mean over the entire unimodal hierarchy, the generations should only reflect variations over g. When varying z, the generations should be complemented by variations from the unimodal hierarchy. Fig. 7 qualitatively demonstrates that image variations are encoded at different levels of abstractions. Tab. 3 quantifies that most measures deteriorate given less variation in the hierarchy.
>
> **“The authors mentioned that they found that increasing the unimodal latent sizes tends to decrease crossmodal coherence. It would be interesting to show this result.”**
>
> Thank you for this suggestion. We will investigate this in our future work.
>
> **“evaluation is restricted to dense modalities such as images and text.”**
>
> We are confident that the current experimental results sufficiently validate our core hypothesis on the merit of latent hierarchical representations. Therefore, we leave the evaluation of further modality types to future work.
>
>
> ---
> We again thank Reviewer Gac5 for the feedback!

---

> > ### Comment · Reviewer_Gac5 · 2021-11-29
> > **Thanks for the response**
> >
> > I thank the authors for their detailed response to reviewer feedback.
> > I do not have any other questions/comments.

---

### Official Review · Reviewer_XV1Z · 2021-11-01

**Correctness:** 4
**Technical Novelty And Significance:** 4
**Empirical Novelty And Significance:** Not applicable
**Recommendation:** 6
**Confidence:** 3

**Main Review:**

It is a very interesting paper. The hierarchical multimodal VAE can capture the realistic variations and help the decoder to share features between different modalities.

Questions:

1. How does the number of hierarchical levels pre-defined in real scenarios? The two experiments used L(m) = 2, will that be better to set it as a tuned parameter?

2. The evaluation metrics used in the paper (Fig 4, 5, 6) seem difficult to conclude which method performs better.  Are there any other quantitative metrics other than likelihood can be used?

3. This method is limited to images and text data. Might be an application limitation for other multimodal data.

Minor: In figure 8, all the captions for the subfigure are the same.


**Summary Of The Paper:**

The authors propose a hierarchical multimodal VAE to capture the heterogeneity through latent variables dependent on a shared top-level variable. CUB and Oxford flower datasets were used for performance evaluation.


**Summary Of The Review:**

Overall, I think the idea to consider the hierarchy for the latent variables between shared and modal-specific variations to learn the representation is novel and interesting. But, I do have concerns about this method’s application in real situations of multimodal learning.

---

> ### Author Response · Authors · 2021-11-23
> **Initial response to Reviewer XV1Z**
>
> We thank Reviewer XV1Z for the feedback. We uploaded a revised version of the paper/code that incorporates this feedback. We highlight our changes in our response below.
>
> **“How does the number of hierarchical levels pre-defined in real scenarios? The two experiments used L(m) = 2, will that be better to set it as a tuned parameter?”**
>
> We agree that the investigation of deeper hierarchies is interesting. The revised paper contains an ablation study with L(1)=1 (MMVAE), L(1)=2, and L(1)=5. We show that performance correlates positively with hierarchical depth across several quantitative measures in a new experiment (Tab. 2). Note that the MVAE attains the best FID and image F1 scores in Tab. 2. However, we find that the model’s qualitative samples lack fidelity (Fig. 7). Furthermore, the model scores exceptionally low on image precision and caption F1 scores.
>
> **“The evaluation metrics used in the paper (Fig 4, 5, 6) seem difficult to conclude which method performs better. Are there any other quantitative metrics other than likelihood can be used?”**
>
> We agree that likelihood estimates alone are not sufficient for a substantive conclusion. Therefore, we included the following measures in the revised paper (Tab. 2):
> FID [1]
> Precision/Recall/F1 [2]
> Average variance of generated features for the image modality
>
> **“This method is limited to images and text data. Might be an application limitation for other multimodal data.”**
>
> Our method is not limited to images and text data (see Section 3). Although we evaluate the proposed method on image and text, the basic idea could also be used on different modalities. For example, consider sound and video modalities, which can also exhibit modality-exclusive variations (see new Fig.1 for terminology) and thereby hierarchical structure. Another example is behavioral and epigenetic data in epidemiology. Note that we are confident that the current experimental results sufficiently validate our core hypothesis on the merit of latent hierarchical representations.
>
> **“Minor: In figure 8, all the captions for the subfigure are the same.”**
>
> Thanks for pointing this out, we corrected this!
>
> ---
> We again thank Reviewer XV1Z for the feedback!
>
> ---
> # References
> ​​[1] M. Heusel, H. Ramsauer, T. Unterthiner, B. Nessler, and S. Hochreiter. Gans trained by a two timescale update rule converge to a local nash equilibrium. NeurIPS 2017
>
> [2] T. Kynka ̈a ̈nniemi, T. Karras, S. Laine, J. Lehtinen, and T. Aila. Improved precision and recall metric for assessing generative models. In NeurIPS, 2019.

---

> > ### Comment · Reviewer_XV1Z · 2021-11-29
> > **Thanks for the response**
> >
> > Dear authors,
> >
> > Thanks for the response. I do not have questions now.

---

### Official Review · Reviewer_j3q4 · 2021-11-02

**Correctness:** 3
**Technical Novelty And Significance:** 2
**Empirical Novelty And Significance:** 2
**Recommendation:** 5
**Confidence:** 4

**Details Of Ethics Concerns:**

N/A.

**Main Review:**

Strengths:
1. The proposed method is interesting and well-motivated.
2. Experimental results are quite comprehensive, and results are promising.
3. The paper is generally well-written and clear.

Weaknesses:
1. The model hinges on the motivation that 'modality-specific variations can sometimes depend on the structure shared across modalities'. Intuitively this seems contradictory since 'modality-specific variations' in one modality are by definition independent of those in another modality, so I am not sure what it means to 'depend on the structure shared across modalities'. It would be good to formally state this assumption and test it on real-world datasets, perhaps using some information-theoretic/dependency-measure metric?
2. The results are better but the improvement could be confounded by the increased number of parameters (I believe the hierarchical model does have more params + potentially more flexibility in modeling the multimodal data). It would be important to control this more carefully when reporting results.
3. Could have some qualitative results + human evaluation results for evaluating caption generation (figure 5).
4. Would also be good to have some human evaluation results for text to image generation (figure 6).
5. The paper tries very hard to distinguish itself from other factorized or hierarchical multimodal generative models (figure 7). While I believe that there is merit to their approach, there are still many possible confounding factors in my opinion, especially regarding the issue of the number of parameters. Also, since each directional arrow in the graphical model is parametrized by multi-layer neural nets, it is not clear the exact difference between explicitly defining a hierarchy of latent variables versus defining a shallow LVM (figure 7c) but having deeper layers in the model.
6. I still have concerns over the novelty of the approach since the main contribution is to define a hierarchy of latent variables. It is not clear when and why this works (related to weakness point 1) so the paper can be improved if there were deeper insights in this part.
7. Quality of figure 8 text can be improved.

**Summary Of The Paper:**

This paper proposes a new type of model called a hierarchical multimodal VAE (HMVAE) that captures modality-specific variations using latent variables dependent on a shared top-level variable, in a manner similar to a multi-layer hierarchy. Their assumption is that modality-specific variations can sometimes depend on the structure shared across modalities which motivates their design decision to have modality-specific variables dependent on a shared top-level multimodal variable, which is in contrast to existing works on multimodal generative models that factorize into marginally independent latent variables to capture modality-specific variations (in other words not depending on a shared top-level multimodal variable).

Experimental results show promising performance on the CUB and the Oxford Flower datasets and outperform existing methods in sample generation quality and quantitative measures as the held-out log-likelihood.

**Summary Of The Review:**

The paper is well-written and the results are promising, but there are concerns over the motivation and novelty as compared to existing work, as well as possible confounding factors in the experimental setup.

---

> ### Author Response · Authors · 2021-11-23
> **Initial response to Reviewer j3q4 (1/2)**
>
> We thank Reviewer j3q4 for the feedback. We uploaded a revised version of the paper/code that incorporates this feedback. We highlight our changes in our response below.
>
> **“The model hinges on the motivation that 'modality-specific variations can sometimes depend on the structure shared across modalities'. Intuitively this seems contradictory since 'modality-specific variations' in one modality are by definition independent of those in another modality, so I am not sure what it means to 'depend on the structure shared across modalities'. It would be good to formally state this assumption and test it on real-world datasets, perhaps using some information-theoretic/dependency-measure metric?”**
>
> The revised paper provides a clearer definition of terminology (Fig. 1), e.g., what we mean by “modality-specific/unimodal variations”. We agree that modality-exclusive variations are independent across modalities (Fig. 1c). However, we state that variations specific to modality $x_i$ are sometimes dependent on shared structure. Note that shared structure cannot contain information beyond $x_i$ -- by definition.
>
> Such dependencies can be found in the CUB and Oxford Flower datasets. First, the captions focus on the object of focus (bird or flower) -- and not on the background. Second, information on writing style cannot be present in the images. The revised paper now contains an explicit example in Fig. 1. Furthermore, our extensive experimental evaluation includes many data samples that substantiate our statement (Sections 4, A).
>
> **“The results are better but the improvement could be confounded by the increased number of parameters (I believe the hierarchical model does have more params + potentially more flexibility in modeling the multimodal data). It would be important to control this more carefully when reporting results.”**
>
> The HMVAE (with a 5-layer image hierarchy) uses the fewest parameters across all models for the experiments in Section 4.3. Note that we merged Sections 4.3 and 4.4, i.e., we now subsume all image-generation experiments in Section 4.3. We added Tab. 4 that shows the number of parameters for the models employed in this new Section.
>
> **“I still have concerns over the novelty of the approach since the main contribution is to define a hierarchy of latent variables. It is not clear when and why this works (related to weakness point 1) so the paper can be improved if there were deeper insights in this part.”**
>
> **“Also, since each directional arrow in the graphical model is parametrized by multi-layer neural nets, it is not clear the exact difference between explicitly defining a hierarchy of latent variables versus defining a shallow LVM (figure 7c) but having deeper layers in the model.**
>
>
> We argue that a shallow LVM performing on-par with a suitably structured model is theoretically learnable but practically challenging. It may require a large model with disproportional capacity, which could generalize poorly, is challenging to train, or requires abundant data.
>
> In contrast, we suggest that a hierarchical latent representation is an inductive bias that captures realistic data variations (Fig. 1) and guides learning. A hierarchical inference model (Fig. 3b) contains edges between latent variables. This is not true for a non-hierarchical network (Fig. 2c). These edges help to decompose the data across multiple levels of abstraction. In contrast to the HMVAE, we also propose deeper hierarchies. These can be beneficial as they allow factorizing variations across more levels of abstractions. We included further respective intuition in Section 3 and Fig. 3-4.
>
> Tab. 2 provides new results (Precision/Recall/F1, FID, sample variance) that substantiate improved performance relative to all baselines. Fig. 7 displays a new qualitative comparison where our model improves semantic coherence. Tab. 2 now provides an ablation between the HMVAE with L(1)=1 (MMVAE), L(1)=2, and L(1)=5. We find that hierarchical depth correlates positively with performance.
>
> For the deep model, we also added experiments where we evaluate hierarchical decomposition. To this end, we first sample conventionally from q(g|x_i). We then sample either conventionally or the mean from any lower-level latent variable z. When sampling the mean over the entire unimodal hierarchy, the generations should only reflect variations over g. When varying z, the generations should be complemented by variations from the unimodal hierarchy. Fig. 7 qualitatively demonstrates that image variations are encoded at different levels of abstractions. Tab. 3 quantifies that most measures deteriorate given less variation in the hierarchy.
>
> ---
>
> Note that our response continues [above (link)](https://openreview.net/forum?id=4V4TZG7i7L_&noteId=0Rik6LO6Cxi).

---

> ### Author Response · Authors · 2021-11-23
> **Initial response to Reviewer j3q4 (2/2)**
>
> **“Could have some qualitative results + human evaluation results for evaluating caption generation (figure 5).”**
> **“Would also be good to have some human evaluation results for text to image generation (figure 6).”**
>
> The results in Fig. 5 of the submitted paper were qualitative. Note that we follow common practice in the multimodal learning literature [1-5] and utilize caption features (Section 4.1).
>
> We assume that “human evaluation results” refers to crowdsourcing, e.g., using Amazon MechanicalTurk to obtain large-scale human evaluations for the qualitative results. We expect that our quantitative measures such as Frechet Inception Distances can also correlate well with human judgment as explicitly shown by [6]. Furthermore, the revised paper contains additional quantitative measures in Tab. 2 (Precision/Recall/F1 across both modalities, sample variance for the image modality). Note that the MVAE attains the best FID and image F1 scores in Tab. 2. However, we find that the model’s qualitative samples lack fidelity (Fig. 7). Furthermore, the model scores exceptionally low on image precision and caption F1 scores.
>
>
> **“Quality of figure 8 text can be improved.”**
>
> Thanks, the revised paper incorporates an improved version.
>
>
> ---
> We again thank Reviewer j3q4 for the feedback. We hope to have appropriately addressed your feedback towards an improved rating.
>
> ---
> # References
>
> [1] Y. Xian, T. Lorenz, B. Schiele, and Z. Akata. Feature generating networks for zero-shot learning. CVPR 2018
>
> [2] M. B. Sariyildiz and R. G. Cinbis. Gradient matching generative networks for zero-shot learning. CVPR 2019
>
> [3] E. Schonfeld, S. Ebrahimi, S. Sinha, T. Darrell, and Z. Akata. Generalized zero-and few-shot learn- ing via aligned variational autoencoders. CVPR 2019.
>
> [4] Y. Shi, N. Siddharth, B. Paige, and P. Torr. Variational mixture-of-experts autoencoders for multi- modal deep generative models. NeurIPS 2019.
>
> [5] Y. Shi, B. Paige, P. H. Torr, and N. Siddharth. Relating by contrasting: A data-efficient framework for multimodal generative models. ICLR 2021
>
> ​​[6] M. Heusel, H. Ramsauer, T. Unterthiner, B. Nessler, and S. Hochreiter. Gans trained by a two timescale update rule converge to a local nash equilibrium. NeurIPS 2017

---

### Official Review · Reviewer_vGFa · 2021-11-03

**Correctness:** 3
**Technical Novelty And Significance:** 2
**Empirical Novelty And Significance:** 1
**Recommendation:** 5
**Confidence:** 4

**Main Review:**

The main contribution of the paper compared to the most similar baseline model MHVAE is in that they utilize the mixture of experts and can, potentially, build a deeper hierarchy.

However, there are a few concerns about the paper, the approach proposed, and the evaluation:

* On page 3, authors clarify the differences between MHAVE and HMAVE(the proposed model). However, MHVAE is also a hierarchical model, and thus, Fig.17a seems not a proper graph for MHVAE. It is not clear why the authors argue that MHVAE is “non” hierarchical model.

* In section 4, the baseline models are not consistent throughout the experiments. Though section 4.4 shows comparison to SOTA methods that have modality specific latent factors, the authors use MVAE and MMVAE in section 4.2 and 4.3, both of which have only the shared latent space without the modality specific space. I find this unfair.

* In section 4.4, the authors implement the MDVAE model by themselves rather than use one of the mentioned MDVAE model implementations. For fair comparison, it would be better to use the baseline network models or justify why that was not done.

* The top row of page 8, they claim “when using deeper hierarchies because a hierarchical inference network can explicitly capture dependencies between different levels of abstraction”. This statement is not substantiated and needs to be grounded in experiments such as the ablation study wrt the hierarchy depth.

* In Figure 6, unconditioned generation results in the first row are provided with the captions above them, inducing confusion (ie, imply conditioning, not unconditional generation). In the unconditioned case, shouldn’t g follow the prior distribution, and not be conditioned on the input, as stated in the text?  This needs clarification.

* In Figure 8 caption, all descriptions indicate (a).


**Summary Of The Paper:**

The authors propose a Hierarchical framework for multimodal learning HMVAE. They define modality specific latent factor as well as the shared latent factor across multiple modalities. They represent modality-specific variations using latent variables dependent on the shared top-level variable. They parameterize the posterior distribution over the shared latent variable using a mixture of experts. The modality specific latent factors are adaptive inferred with both bottom-up and top-down information. They evaluate the proposed method on the Oxford Flower and the CUB datasets with various cross-modal experiments.


**Summary Of The Review:**

Two primary element should be resolved to provide convincing arguments about the utility of the proposed approach. (1) Consistent baseline model comparisons throughout section 4 experiments.  (2) Provide clear comparative analysis with MHVAE and show what is specifically novel (or identical to) this close competitor and substantiate improved performance in clear experiments.

---

> ### Author Response · Authors · 2021-11-15
> **Initial response to Reviewer vGFa**
>
> Thank you for your insightful comments. We will provide a complete response addressing all your comments and suggestions with an updated paper by Nov. 22. However, we would like to clarify one aspect at this point:
>
> > “In section 4.4, the authors implement the MDVAE model by themselves rather than use one of the mentioned MDVAE model implementations. For fair comparison, it would be better to use the baseline network models or justify why that was not done.”
>
> One of the reasons why we reimplemented the MDVAE baseline (details in Section 4.4) is that we could not find any publicly available MDVAE implementation that can cope with the data types employed in our work.  Could you please suggest any open-source implementation that we should use for a fair comparison in our experiments?

---

> > ### Author Response · Authors · 2021-11-23
> > **Response to Reviewer vGFa (1/2)**
> >
> > We again thank Reviewer vGFa for the feedback. We uploaded a revised version of the paper/code that incorporates this feedback. We highlight our changes in our response below.
> >
> > **“Provide clear comparative analysis with MHVAE and show what is specifically novel (or identical to) this close competitor and substantiate improved performance in clear experiments”**
> >
> > **“Provide clear comparative analysis with MHVAE [...] and substantiate improved performance in clear experiments”**
> >
> > There are five differences between the MHVAE and our model:
> > We support arbitrary hierarchical depth, the MHVAE supports two levels.
> > We use a hierarchical inference network, the MHVAE does not.
> > We use multimodal posteriors via top-down inference, the MHVAE uses unimodal posteriors over z.
> > We parameterize q(g|x) as a mixture of experts, the MHVAE uses “domain dropout”.
> > We consider modalities with many variations in both modalities.
> >
> > The revised paper improves the comparative analysis (Section 2). Section 3 and Figure 4 provide further intuition on why deeper hierarchies are helpful.
> >
> > Tab. 2 provides new results (Precision/Recall/F1, FID, sample variance) that substantiate improved performance across all baselines. Fig. 7 displays improved semantic coherence in a qualitative fashion. Note that the MVAE attains the best FID and image F1 scores in Tab. 2. However, we find that the model’s qualitative samples lack fidelity (Fig. 7). Furthermore, the model scores exceptionally low on image precision and caption F1 scores.
> >
> > Note that our work and the MHVAE paper were developed independently. To our knowledge, we are the first work that explicitly demonstrates multimodal hierarchical decomposition in multimodal VAEs (Fig. 7, Tab. 2).
> >
> > **“The top row of page 8, they claim “when using deeper hierarchies because a hierarchical inference network can explicitly capture dependencies between different levels of abstraction”. This statement is not substantiated and needs to be grounded in experiments such as the ablation study wrt the hierarchy depth.”**
> >
> > Tab. 2 contains new results: an ablation between the HMVAE with L(1)=1 (MMVAE), L(1)=2, and L(1)=5. We find that the more hierarchical depth substantially improves several quantitative measures in terms of coherence and heterogeneity.
> >
> > For the 5-layer HMVAE, we also provide new results on hierarchical decomposition. To this end, we perform an experiment where we first sample conventionally from q(g|x_i). We then sample either conventionally or the mean from any lower-level latent variable z. When sampling the mean over the entire unimodal hierarchy, the generations should only reflect variations over g. When varying z, the generations should be complemented by variations from the unimodal hierarchy. Fig. 7 qualitatively demonstrates that image variations are encoded at different levels of abstractions. Tab. 3 quantifies that most measures deteriorate given less variation in the hierarchy. These results again prove that hierarchical depth is essential for improved generative performance.
> >
> > **“On page 3, authors clarify the differences between MHAVE and HMAVE(the proposed model). However, MHVAE is also a hierarchical model, and thus, Fig.17a seems not a proper graph for MHVAE. It is not clear why the authors argue that MHVAE is “non” hierarchical model.”**
> >
> > We agree that the MHVAE [1] is a hierarchical model because its generative model is hierarchical (as in Fig. 3a, but with two levels). However, its inference model is non-hierarchical (Eq. 9 from the MHVAE paper) as shown in Fig. 2c and Eq. 19. There is no edge between z_i and g for any i. We made this point clearer in the revised paper.
> >
> > **“In section 4, the baseline models are not consistent throughout the experiments. Though section 4.4 shows comparison to SOTA methods that have modality specific latent factors, the authors use MVAE and MMVAE in section 4.2 and 4.3, both of which have only the shared latent space without the modality specific space. I find this unfair.”**
> >
> > The revised paper merges Sections 4.3 and 4.4 into a single Section 4.3. Within this section, we now compare to all baselines. We also provide a clearer as well as more exhaustive evaluation and introduce new quantitative measures. Note that we have limited time and computational resources. Therefore, we must prioritize and focus on the most important experiments.
> >
> > ---
> >
> > Note that our response continues [above (link)](https://openreview.net/forum?id=4V4TZG7i7L_&noteId=IXmS7IlqIVO).

---

> > ### Author Response · Authors · 2021-11-23
> > **Response to Reviewer vGFa (2/2)**
> >
> > **“In section 4.4, the authors implement the MDVAE model by themselves rather than use one of the mentioned MDVAE model implementations. For fair comparison, it would be better to use the baseline network models or justify why that was not done.”**
> >
> > One of the reasons why we reimplemented the MDVAE baseline (details in Section C.3) is that we could not find any publicly available MDVAE implementation that can cope with the data types employed in our work.  Since we had no suggestion for implementations (after our question on November 15), we are still not aware of any suitable open-source implementation.
> >
> > **“In Figure 6, unconditioned generation results in the first row are provided with the captions above them, inducing confusion (ie, imply conditioning, not unconditional generation). In the unconditioned case, shouldn’t g follow the prior distribution, and not be conditioned on the input, as stated in the text? This needs clarification.”**
> >
> > The caption was indeed incorrect. Both results represented conditional generations in the submitted paper. Thanks for pointing this out!
> >
> > **“In Figure 8 caption, all descriptions indicate (a).”**
> > Thanks! The bullet points should have increased from (a) to (d) in the submitted paper.
> >
> >
> > ---
> >
> > We again thank Reviewer vGFa for the feedback. We hope to have appropriately addressed your feedback towards an improved rating.
> >
> >
> > ---
> > # References
> > [1] M. Vasco, F. S. Melo, and A. Paiva. Mhvae: a human-inspired deep hierarchical generative model for multimodal representation learning. arXiv 2020.

---

### Decision · Program_Chairs · 2022-01-20

**Decision:**

Reject

**Comment:**

PAPER: This paper presents a multimodal auto-encoder architecture built on the premise that unimodal variations can be best generated when taking advantage of a shared latent space. This is operationalized by defining a hierarchical model with two primary levels: a shared structure space and unimodal variations (which could be multi-layer).
DISCUSSION: The reviewers and follow-up discussion brought many questions and issues. The authors submitted a significantly revised version of their paper which clarified many issues and added a few extra results. While many of the reviewers’ questions were addressed by the authors, it seems that reviewers ended up not changing significantly their review scores. One fundamental concern is if the basic assumption about the shared structure is effectively the proper way to approach such generative modeling task. The experimental for image generation did not seem to support this hypothesis.
SUMMARY: While the revised version was an improvement over the original submission, improving clarity and adding some experimental measures, the experimental results did not seem to always support the main hypothesis. Human evaluation results may help in this direction.